# PoreDesigner for tuning solute selectivity in a robust and highly permeable outer membrane pore

Ratul Chowdhury [1], Tingwei Ren[1], Manish Shankla[2], Karl Decker[2], Matthew Grisewood[1], Jeevan Prabhakar[1], Carol Baker[3], John H. Golbeck[3,4], Aleksei Aksimentiev [2], Manish Kumar[1] & Costas D. Maranas [1]

Monodispersed angstrom-size pores embedded in a suitable matrix are promising for highly selective membrane-based separations. They can provide substantial energy savings in water treatment and small molecule bioseparations. Such pores present as membrane proteins (chiefly aquaporin-based) are commonplace in biological membranes but difficult to implement in synthetic industrial membranes and have modest selectivity without tunable selectivity. Here we present PoreDesigner, a design workflow to redesign the robust beta-barrel Outer Membrane Protein F as a scaffold to access three specific pore designs that exclude solutes larger than sucrose (>360 Da), glucose (>180 Da), and salt (>58 Da) respectively. PoreDesigner also enables us to design any specified pore size (spanning 3–10 Å), engineer its pore profile, and chemistry. These redesigned pores may be ideal for conducting sub-nm aqueous separations with permeabilities exceeding those of classical biological water channels, aquaporins, by more than an order of magnitude at over 10 billion water molecules per channel per second.

[1] Department of Chemical Engineering, The Pennsylvania State University, University Park, PA 16802, USA. [2] Department of Physics, University of Illinois at Urbana-Champaign, Urbana, IL 61801, USA. [3] Department of Biochemistry and Molecular Biology, The Pennsylvania State University, University Park, PA 16802, USA. [4] Department of Chemistry, The Pennsylvania State University, University Park, PA 16802, USA. These authors contributed equally: Ratul Chowdhury, Tingwei Ren. Correspondence and requests for materials should be addressed to M.K. (email: manish.kumar@psu.edu) or to C.D.M. (email: costas@psu.edu)

Precise chemical separations such as desalination and distillation are among the most challenging and resource-intensive industrial process operations practiced today with an annual energy consumption of ~ 50 Quads ($50 \times 10^{15}$ BTUs) in the United States alone[1]. Membranes are generally defined as thin, selective barriers that ideally only allow select molecules to permeate through while rejecting others[2]. Permeability and selectivity are key metrics of performance for membrane separations. For biological membranes a nondimensional driving force, defined as the osmotic gradient times the molar volume of water, is used in defining[3] permeability as the fluid flux per unit driving force. Membrane selectivity is consequently defined as the ratio between the permeability of two solutes, two solvents or between a solute and a solvent. Membrane separation offers advantages such as higher selectivity, simpler operation, and higher compactness over other (in many cases thermally driven) separation processes[4]. They are increasingly being applied to a number of industrial sectors including water treatment[5] industrial gas separations[6], $CO_2$ capture[7], food processing[8], and bio-pharmaceutical separations[9]. A variety of materials such as synthetic polymers[10], ceramics[11], metals[12], and cellulose[13] can be used to synthesize membranes. However, several challenges still remain in membrane materials design, particularly at the pore scale and translation of such designs to large areas necessary for application[14,15]. These challenges include: (1) overcoming the trade-off between selectivity and permeability to develop membranes with high selectivity and permeability because improvements in selectivity would simplify multiple separation steps and decrease separation costs significantly; (2) designing angstrom-scale pores that result in the same angstrom-scale separations in synthesized membranes. This would be critical for the efficient separation of small molecules such as ions, gases, and small organics; and (3) synthesizing membranes with uniform pore size distributions. The elimination of polydispersity in membrane pore size would greatly enhance selectivity performance[15].

To meet these criteria, in recent years, new materials including zeolites[16], carbon nanotubes[17], graphene oxide[18], and membrane proteins[19], and membrane protein mimics[20] have emerged as advanced membrane materials to assemble the desired pore geometry. However, designing sub-nm pores with perfectly monodisperse distributions is still an unmet challenge and no procedure exists to rationally design the continuum of pore sizes between 3 and 10 Å. Membrane protein channels and biomimetic membranes based on these proteins provide the possibility of realizing sub-nm pore size membranes with perfectly monodisperse pore size distributions, retaining high selectivity while maintaining high permeability[15,21]. A well-known example of such membranes uses water transport membrane proteins, aquaporins (AQPs), incorporated into liposomes and further stabilized in polyamide polymer membranes for water desalination[19]. Membrane protein redesign studies have been performed and reported on gated ion channels[22–25] and helix-bundle ion transporters[26]. In a recent study, Liu et al.[27] engineered a ferrichrome outer membrane iron transporter (FhuA) to attain pore sizes of $1.6-2.7$ nm and explored its transport properties.

AQP1 is the most studied AQP isoform, so we chose it as a model water channel for our study. AQP1 pore has an hour-glass structure with a constriction diameter of ~2.7 Å. Rapid and selective water permeation through AQPs at ~3 billion water molecules/channel/s, makes them ideal for desalination. However, the advantages of AQPs for high permeability and high selectivity membrane applications are still being debated due to questions regarding the long-term stability of this alpha helical protein[28], the low density of proteins embedded in membranes[29], and its pore wall chemistry which forms hydrogen bonds with the permeating water[30] molecules that may impede single-channel osmotic permeability. Further, AQP-based membranes do not allow for selective removal of larger solutes in the sub-nm range.

To this end, we put forth a predictive platform to computationally redesign the pore-constricting residues of a highly stable outer membrane protein[31] of β-barrel family of channels present in bacteria[32]. These proteins have been extensively engineered and shown to be stable under varying temperature and chemical conditions[33,34]. In particular, we worked with the trimeric *Escherichia coli* protein outer membrane porin type F (OmpF, wild-type (WT) pore size of ~ 11 Å) to attain desired pore sizes that could enable precise molecular separations. OmpF[35] is mainly involved in transporting ions[36], antibiotics[37], small sugars[38], polyamines[38], and amino acids[38]. OmpF also has an hour-glass-shaped structure, similar to that of AQP1, but the pore constriction diameter of OmpF is ~4 times larger. The stability and mutation tolerance of OmpF[39,40] makes it a suitable candidate for computational redesign and subsequent experimental validation for performing separations at the angstrom scale. In addition, it can be easily assembled into stable two-dimensional crystals formed within block-copolymer membrane matrices[41] which may be ideal for preparation of larger scale separation membranes[19].

In this work, we outline the systematic workflow of PoreDesigner for a predictive platform to utilize the OmpF scaffold to design angstrom-scale pore sizes with specific solute selectivity and high osmotic permeabilities. This was accomplished by leveraging the protein design algorithm IPRO (Iterative Protein Redesign and Optimization suite of programs)[42] and subsequent application of molecular dynamics (MD) simulations and validation using stopped-flow light scattering experiments. The core computational module of PoreDesigner restricts the modification of the pore constriction residues to long side-chain and hydrophobic amino acids and identifies an optimal set of rotational isomers (from a rotamer library) for the altered residues that avoid backbone and side-chain clashes using a mixed-integer linear optimization program (MILP). Long side-chain hydrophobic amino acids were selected with the dual objective of obtaining smaller pores with hydrophobic side chains that extend into the pore lumen to provide selectivity while maintaining high osmotic permeability based on the hypothesis that reducing water−pore wall interactions will lead to increased permeability[43,44]. Designs identified by the MILP problem were retained so long as (1) they minimized water wire to pore wall interactions, or (2) had pore sizes smaller than the desired size, using an interaction energy calculation check and a pore area estimation criterion. Structural investigation on the designs revealed three distinct topologies of pore geometries: (a) uniform pore closure designs (UCD) with a smaller but nearly cocentric pore eyelet diameter resulting from an orderly distribution of similar-side-chain size hydrophobic amino acids along the pore perimeter, (b) off-center pore closure designs (OCD) which involve a pore center that is displaced towards the perimeter compared to the WT OmpF pore utilizing long side-chain hydrophobic resides on one side of the pore and smaller ones on the other, and (c) cork-screw designs (CSD) which introduce a lateral twist as we proceed along the pore axis stemming from alternating stacking long with short side-chain amino acids (Fig. 1). We were able to redesign the $7 \times 11$ Å OmpF WT pore to obtain an array of designs with varying pore size profiles sampling pore sizes across the 3–10 Å range and experimentally tested a subset of these designs in the 3–4 Å range, critical for the most challenging separations. The permeabilities of tested designed ranged from 1.0 ($\pm0.23$)$\times10^{-12}$ $cm^3$/s for the WT to 4.4 ($\pm0.93$)$\times10^{-12}$ $cm^3$/s (the ±values are standard deviations based on measurements from three replicates in both WT and CSD) for the selected CSD design compared to ~$9\times10^{-14}$ $cm^3$/s estimated for AQP1[45]. CSD osmotic water permeabilities

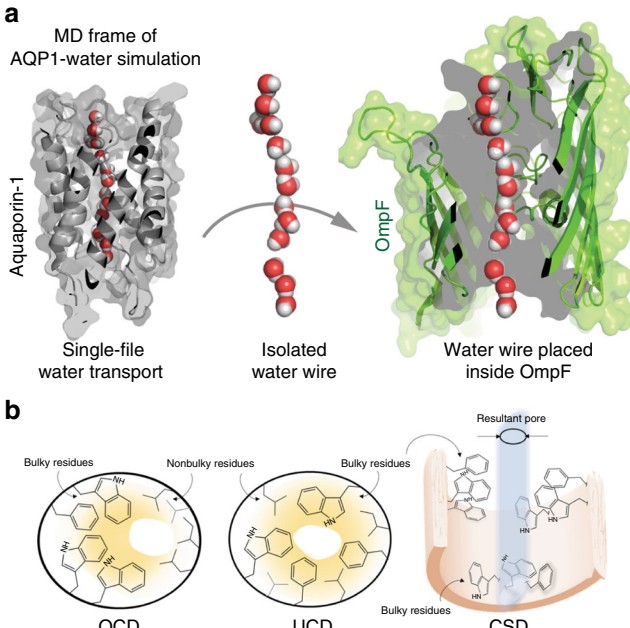

**Fig. 1** Water wire from Aquaporin 1 (AQP1) was used as a template to redesign OmpF pore geometry. **a** The left panel shows a frame from an MD simulation of single-file water permeation through AQP1. The pore wall residues capable of forming hydrogen bonds with the permeating water wire have been highlighted in yellow. The water wire is isolated with its geometry preserved and is thereafter placed in the OmpF pore. The pore-constricting residues are altered such that they fill up the space around the water wire forming a molecular mold of the selective internal geometry of AQP1 within OmpF beta scaffold. **b** The three distinct internal pore geometries of OmpF that resulted from the employed redesign procedure included: (i) off-center pore closure design (OCD), (ii) uniform pore closure design (UCD), and (iii) cork-screw design (CSD)

were not only higher than AQP1 by an order of magnitude but were also an order of magnitude higher than the highest reported permeabilities of any channel of the aquaporin family of proteins ($2.4$ $(\pm 0.47) \times 10^{-13}$ cm$^3$/s determined for *Rhodobacter sphaeroides* AqpZ[29,46]). Solute rejection capabilities of these channels were evaluated using stopped-flow experiments with various solutes. These results demonstrate the potential of computationally tuning the pore diameter of OmpF in response to desired separations.

## Results

**Recapitulating water wire geometry of AQP1 in OmpF mutants**. Aquaporins have the ideal internal pore geometry for selective and highly permeable water channels but the pore wall interacts strongly with the permeating water wire[47], indicating the possibility of enhancing permeability at similar size ranges without sacrificing selectivity. All aquaporins have a conserved asparagine-proline-alanine motif (known as the NPA motif) near the constriction region, in which the Asn interacts with the water wire by forming hydrogen bonds. These interactions impede the hydraulic permeability through AQP1. A recent study[47] showed that in addition to the NPA motif there are 12 amino acids along the internal pore profile of AQP1 that can form hydrogen bonds with the water wire. Further, the number of hydrogen bonds between the water wire and the inner pore wall of AQPs was directly related to the single-channel permeability of the pore. Our aim is to redesign the water channel such that it minimizes interaction with the permeating water wire, thus eliminating

hydrogen bonds in the central part of the channel but retaining the water wire geometry. To discern the unique water wire configurations (Supplementary Figure 1 and Supplementary Table 1) through AQP1, we examined individual frames of AQP1-water using all-atom 10 ns MD simulations (Fig. 1). Such water wires were subsequently positioned inside the OmpF pore and PoreDesigner was used to alter the pore-constricting residues to form the equivalent of a molecular mold around the water wire (Fig. 1).

**Categorization of redesigned pores**. Water wires were placed inside the WT OmpF (2omf.pdb) pore and used as input for PoreDesigner. The pore-constricting residues were altered to fill up the annular space using hydrophobic, long side-chain amino acids with the objective of designing a narrow yet hydrophobic pore with minimal water−wire interaction. An explicit constraint was imposed to ensure that the distance between any hetero-atom of the pore constriction residue and the water wire oxygen was greater than the sum of their van der Waals radii to preclude the possibility of arriving at designs that occlude the pore. PoreDesigner reduces binding with the central water wire by maintaining their respective interaction energies at their maximum by replacing the pore constriction residues of WT OmpF with long side-chain, hydrophobic amino acids such as tryptophan (Trp), phenylalanine (Phe), and tyrosine (Tyr).

PoreDesigner yielded 40 different OmpF designs with pore sizes less than 4 Å. Analysis of the engineered designs revealed that they conform to three categories (Fig. 1) based on the resultant internal pore geometry: (1) only Trp/Phe mutations resulting in a narrower but off-center pore lumen (OCD: off-center pore closure design), (2) a smaller cocentric pore with the bulky groups (such as Trp, Phe, and Tyr) interspersed with less bulky alanines and valines arranged in a single plane (UCD: uniform pore closure design), and (3) regularly patterned larger with smaller side-groups along the pore profile resulting in an internal pore geometry that involves a twist (cork-screw designs —CSDs). There were two off-center (OCD) designs for which we allowed the mutation of the 25 pore constriction residues to: (a) only Phe mutations (TFPhe), and (b) only Trp mutations (TFTrp). A biased distribution of bulky groups (Phe/Trps) towards one side of the pore periphery resulted in a smaller pore with its center away from the large residues. The presence of 25 Phe/Trps led to steric clashes forcing most of the Phe/Trps side chains to face away from the pore lumen (Fig. 2) thus enhancing pore wall hydrophobicity only. In the remaining 38 designs, all hydrophobic long side-chain amino acids were permitted, thus enabling them to sample smaller pore sizes by placing long side-chain residues interspersed with short amino acids (generally alanines, valines, and leucines). These designs (UCD) alleviate the possibility of a Phe-Phe/Trp-Trp side-chain steric clashes akin to the OCD designs (Fig. 1). We identified 31 UCD and seven CSD designs. We chose the smallest predicted pore size design from each type for subsequent osmotic permeability and solute rejection experiments and molecular dynamics simulations. The predicted pore constriction dimensions after MD simulations for the three selected designs were: $3.54 \times 3.25$ Å, $3.18 \times 3.12$ Å, and $3.05 \times 3.01$ Å for OCD, CSD, and UCD protein designs, respectively. These OmpF mutant proteins and the WT OmpF proteins were produced by expression from synthetic genes cloned into the pET23a(+) expression plasmid vector transformed into *E. coli* BL21(DE3) Omp8 Rosetta (ΔlamBompF::Tn5 ΔompAΔompC) mutant strain. The purified proteins were incorporated into liposomes for assessment of single-channel water permeability and solute passage as described in subsequent sections.

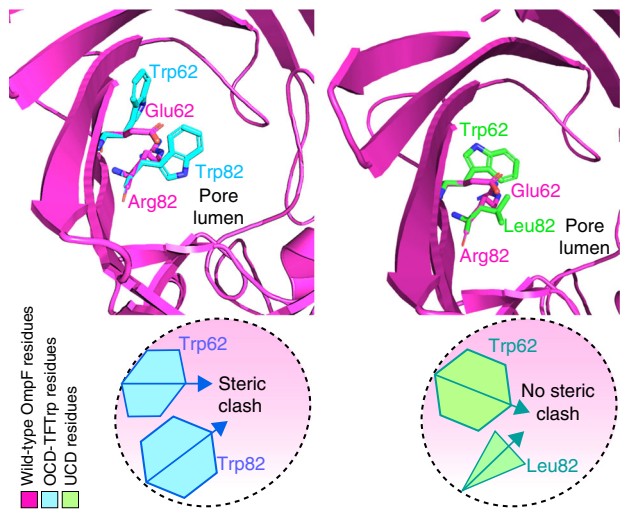

**Fig. 2** OCD-TFTrp design shows steric clash but UCD does not. The OCD_TFTrp design has adjacent tryptophans clashing (left) resulting in some of the side chains facing away from the pore lumen, thereby yielding pore sizes larger than expected. However, in a UCD design (right), an R82L mutation alleviates a steric clash with Trp62 (unlike OCD-TFTrp). UCD designs are seen to intersperse smaller side-chain hydrophobic amino acids between longer ones so their side chains face the pore lumen resulting in smaller pore sizes

Before testing the three designs for water transport, the Kyte−Doolittle hydrophobicity scores of the three designs were calculated (Supplementary Methods) and contrasted with that of WT OmpF (Table 1). The relative order of inner pore wall hydrophobicities was seen to be CSD > OCD > UCD > wild type OmpF. We hypothesized that the experimentally measured single-channel permeabilities will follow the same trend as increasing hydrophobicity based on our design principle of eliminating water−pore wall interactions to enhance permeability.

The hydrophobicity trends reveal that the OCD-TFTrp mutant has the highest estimated outer pore wall hydrophobicity. This is possibly due to the steric clashes between contiguous tryptophans (Fig. 2) where the majority of the 25 pore-constricting tryptophans are forced away from the lumen.

In addition, we also used PoreDesigner to predict designs that span the remaining 4–10 Å range. The overall goal is to precisely match any desired pore size needed for separations spanning the sub-nm (3–10 Å) range. PoreDesigner was accordingly modified to only accept pore designs with pore constriction diameter between a prespecified range $D^{min}$ and $D^{max}$. For example, setting $D^{min}$ and $D^{max}$ values to 5 and 6 Å respectively yields OmpF designs with pores predicted to be within this range. We identified 17 additional designs (Fig. 3) with at least two designs within a pore size bin of range 1 Å starting from 4 to 9 Å. We used the aforementioned structural classification scheme and developed ten OCD designs, three UCD and four CSD designs. Generally, the smaller the desired pore size, the higher was the number of required mutations. OCD type designs were seen to be most prevalent spanning almost the entire sub-nm range (Fig. 3b). Whereas CSD and UCD type designs were limited to the mid region of the sub-nm spectrum.

**Experimental validation of pore designs**. WT OmpF and three OmpF mutants were chosen to measure their single-channel permeability and solute passage rates experimentally. The three

OmpF mutants that we chose were OCD with in silico estimated pore sizes of 3.25 Å (minor axis of elliptical pore cross-section), CSD with pore size of 3.12 Å, and UCD with estimated pore size of 3.01 Å. The amino acid sequences of these mutant proteins are provided in Supplementary Figure 6.

**Solute rejection of OmpF mutants**. We estimated the approximate molecular weight limit at which solute rejection for WT OmpF and the three OmpF mutants occurred using stopped-flow light scattering experiments (see Fig. 4 for details). For WT OmpF, the light scattering intensity decreased at the second stage when WT OmpF-reconstituted liposomes were exposed to NaCl containing hypertonic solutions. Also, the light scattering intensity decreased at the second stage when WT OmpF-reconstituted liposomes were exposed to glycine, glucose or sucrose containing hypertonic solutions. This indicates that WT OmpF is permeable to these solutes. The light scattering intensity leveled off when exposing the proteoliposomes to PEG600 containing hypertonic solutions (Fig. 4). This observation demonstrated that WT OmpF can reject PEG600 (600 Da) or larger molecules, which is consistent with previous reports[35].

For UCD, the light scattering intensity leveled off when exposing the liposomes to all the solutes used including NaCl (58.5 Da), leading us to conclude that this mutant can substantially reject molecules larger than 58.5 Da (Fig. 4). We also estimated the approximate molecular weight exclusion limit for OCD and CSD (Supplementary Methods and Supplementary Figure 3 for details). Based on the solute rejection experiments above, we estimated the molecular exclusion limit of the three mutants to have the following sequence: WT (~600 Da) > OCD (~342 Da) > CSD (~180 Da) > UCD (~58 Da) (Fig. 4), which follows the same trend as the designed pore sizes. Thus, small molecule separation membranes can be developed by a selection of different pore size OmpF mutants for biomimetic membranes. The mutant with the smallest pore size, UCD, has ionic solute rejection properties similar to aquaporins (while not excluding protons), and can be selected as a candidate protein for developing membrane protein-based biomimetic desalination membranes.

**Single-channel permeability of OmpF mutants**. Recent literature has focused on emphasizing the importance of membrane design efforts that lead to high selectivity while maintaining or increasing current membrane permeabilities[15]. Solute rejection experiment results showed that high molecular selectivity can be achieved by designing OmpF mutants with different pore sizes through the PoreDesigner workflow. In addition to estimating selectivity, we also evaluated OmpF mutant permeabilities, which were specifically characterized by determining single-channel permeability of each mutant.

Figure 4 shows light scattering curves obtained from vesicle membranes with reconstituted WT and mutant OmpF proteins at a lipid to protein mass ratio of 400 (LPR400), with net permeabilities between 2049 and 3411 μm/s. Because net permeability can depend both on the number of proteins reconstituted per vesicle and the single-channel permeability[29], for more accurate comparison between mutants, we calculated the single-channel permeability of WT OmpF and its mutants. By taking the ratio of the number of proteins to the number of vesicles, we can obtain the average number of proteins per vesicle ($N_{pro}/N_{ves}$) (Supplementary Figure 5). Combining vesicle permeability and average number of OmpF proteins per vesicle, we calculated average single-channel permeability of OmpF and its mutants (Fig. 4). CSD had the highest single-channel permeability followed by OCD and UCD, which have similar single-channel

**Table 1 Ranges of values for the inner pore wall, outer pore wall, and overall hydrophobicity scores**

| Designs | Inner pore wall hydrophobicity score[a] $\Delta G_{transfer}^{water \to ethanol}$ (kcal/mol) | Outer pore wall hydrophobicity score $\Delta G_{transfer}^{water \to ethanol}$ (kcal/mol) | Overall hydrophobicity score $\Delta G_{transfer}^{water \to ethanol}$ (kcal/mol) |
|---|---|---|---|
| CSD | −97.7 | −76.5 | −174.2 |
| OCD | −69.2 | −81.2 | −150.4 |
| UCD | −65.6 | −71.3 | −136.9 |
| Wild type | −61.3 | −68.5 | −129.8 |

[a]A higher negative value represents a higher hydrophobicity[58]. This scale reports the free energies of transfer of different amino acid side chains from water phase to ethanol. As a result, the hydrophobic amino acids have a lower free energy of transfer than charged amino acids

**Fig. 3** Twenty OmpF mutants spanning the entire sub-nm range were designed. **a** Plot of the number of mutations vs. pore diameter for 20 mutants (including three mutants that were validated experimentally before MD simulations). The general trend indicates that the smaller the desired pore, the greater the number of mutations required. **b** Plot of the number of designs for each pore size and type classification

permeabilities and all the three mutants have single-channel permeabilities higher than WT OmpF. This serves as the experimental corroboration of the predicted water permeation rates from the inner pore wall hydrophobicities. Compared to aquaporins, single-channel permeabilities of WT OmpF and its mutants are at least an order of magnitude higher[48], the highest single-channel water permeability of OmpF mutants is ~18 times faster than that of the *E. coli* AqpZ, which was measured using the same platform[29] and ~49 times faster than that of AQP1 reported in literature[45].

**Molecular dynamics simulation of the pore designs.** Using the all-atom MD method, we independently assessed osmotic permeability of the WT OmpF and the three experimentally verified designs, starting from the molecular configurations suggested by PoreDesigner. The monomeric proteins were patched to form trimers using a VMD[49] plugin, set in a lipid-bilayer and solvated in 1 M NaCl solution (Fig. 5). Osmotic permeabilities were evaluated (from the rate of vesicle volume changes[50]) and averaged through each monomer of the trimeric molecule and the variabilities during the last 30 ns of the 35 ns simulation were reported (Fig. 5). The MD-computed osmotic permeabilities of OmpF and the three designs are seen to corroborate the same single-channel permeability trend as seen in stopped-flow light scattering experiments. Highest permeability of CSD reaffirms the applicability of using inner pore wall hydrophobicity scores as a surrogate to predict relative channel permeabilities.

The ionic conductances of the WT and mutant pores were evaluated by simulating the systems under a transmembrane voltage of 500 mV (Fig. 5). The ionic conductance was determined by averaging instantaneous displacements of ions over the last 25 ns of the 35 ns MD trajectoires[51]. All the three mutants exhibited negligibly low conductances which were about an order of magnitude lower than that of WT OmpF with the CSD mutant being the most conductive of the three as expected from solute rejection experiments. While solute rejection experiments identified UCD to be more restrictive towards salt compared to OCD, MD simulations, as conducted, do not seem to be conclusive regarding the difference between the two mutants (Fig. 5).

Further analysis of the MD trajectories identified the inner volume of the WT and mutant pores accessible to water (Fig. 5). The volumes were determined by averaging water density over the last 10 ns of the MD simulations carried out under a 500 mV bias. The volume density maps reveal the UCD mutant to have the narrowest pore constriction, which correlates with the best solute rejection performance of that mutant. Figure 5 shows the major pore constriction diameters as measured using PoreDesigner (before MD simulations were performed) along with that observed during the 5 ns of MD. The closely packed side chains of OCD and UCD allowed marginal movement of pore constriction amino acid side chains, thereby showing less variability in the pore size during the course of MD. However, the bulky groups of CSD are stacked in different planes thus allowing some movement of the pore constriction side chains leading to higher variability in pore sizes during water permeation. The same

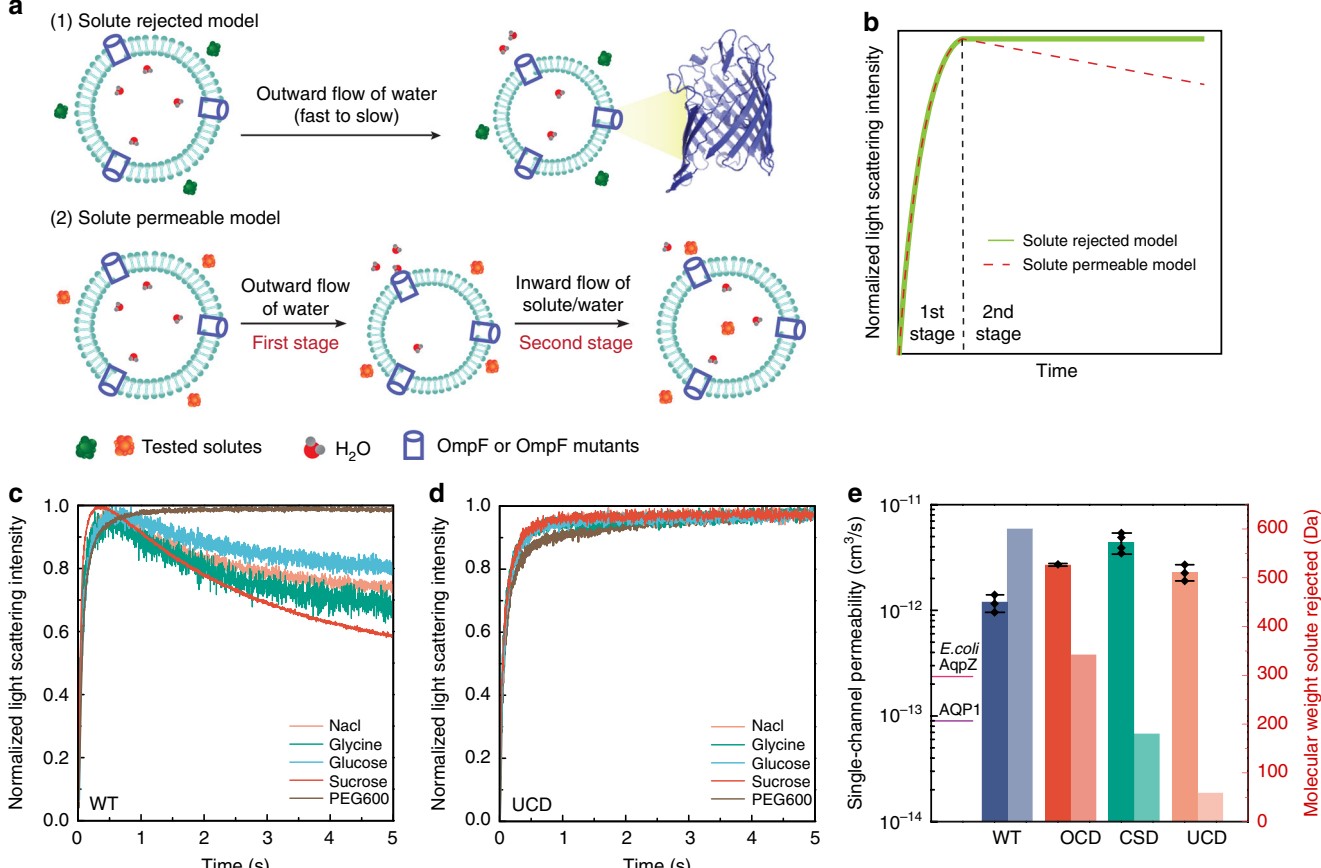

**Fig. 4** Osmotic shock stopped-flow light scattering experiments were used to assess transport properties. Stopped-flow light scattering experiments revealed an order of magnitude or higher permeability than aquaporins for WT OmpF protein and its mutants as well as solute retention trends seen in OmpF protein mutants. **a** When OmpF (or OmpF mutant) containing proteoliposomes are mixed with hypertonic solutions, two different transport models can be observed based on whether the solute is permeable to the porin or not. **b** In the stopped-flow setup, for solute excluded model, normalized light scattering intensity levels off during the second stage as there is no inflow of water and solutes; for solute permeable model, normalized light scattering intensity decreased during the second stage due to inflow of water and solutes. **c** OmpF (WT) rejects PEG600 (600 Da) and larger molecules and thus only the PEG600 curves show no decreasing portion of the curve. **d** UCD rejects NaCl (58.5 Da) and larger molecules as there is no decreasing portion of the stopped-flow curve for any of the solutes tested. **e** Summary of the estimated solute rejection (light bars) and single-channel permeability (dark bars) of OmpF WT and the three OmpF mutants (details in Supplementary Figures 3 and 4). The two y-axes represent permeability (black left y-axis) or the molecular weight cutoff data (red right y-axis). Curves shown in panels **c** and **d** are averages of 6–10 traces from each stopped-flow light scattering experiment. Each experiment was conducted at least three times with independent vesicle preparations (complete data in Supplementary Methods and Supplementary Figure 3)

PoreAnalyzer module (Supplementary Methods) that was used in PoreDesigner was used for assessing the pores from the MD trajectories.

To assess the effect of relative hydrophobicity of the mutant pores, we determined the average number of hydrogen bonds between the OmpF monomer and water located in each of the three regions of the pores (defined in Fig. 5). A hydrogen bond was reported if the water molecule was within 0.3 nm of a protein atom capable of forming a hydrogen bond similar to as explained in Ireta et al.[52] and Durrant et al.[53]. It was also ensured that the protein atom–water hydrogen–water oxygen angle was 20° or less. The number of calculated bonds was averaged over entire 35 ns MD trajectories with error bars showing standard deviations (Fig. 5). The constriction region (Fig. 5) shows progressively decreasing number of hydrogen bonds as the number of mutations (in the constriction region) as the pore is occluded by more hydrophobic residues. This attests to the effectiveness of PoreDesigner's design objective of replacing pore constriction residues with hydrophobic ones to limit pore

wall−water wire interactions in order to arrive at designs with high single-channel permeabilities while tuning size selectivity.

## Discussion

Ultrapermeable membranes (dense solution, diffusion-based or channel-based) have emerged as a promising alternative to energy-intensive separations including desalination and water purification. Here, we have put forth a computational (i.e., PoreDesigner) and experimental workflow that relies on the mechanically, chemically, and mutationally stable beta-barrel scaffold of OmpF as a candidate for synthesizing channel-based membranes for speedy aqueous-phase separations of specific solutes, thus expanding the selectivity range of current AQP-based biomimetic membranes. A simple theoretical analysis of ion conductance targets for desalination membranes based on OmpF channels is described in the Supplementary Methods. This analysis reveals that in order for such membranes to be usable for seawater desalination, the estimated maximum total conductance values is 0.018 nS for seawater desalination (assuming a feed of

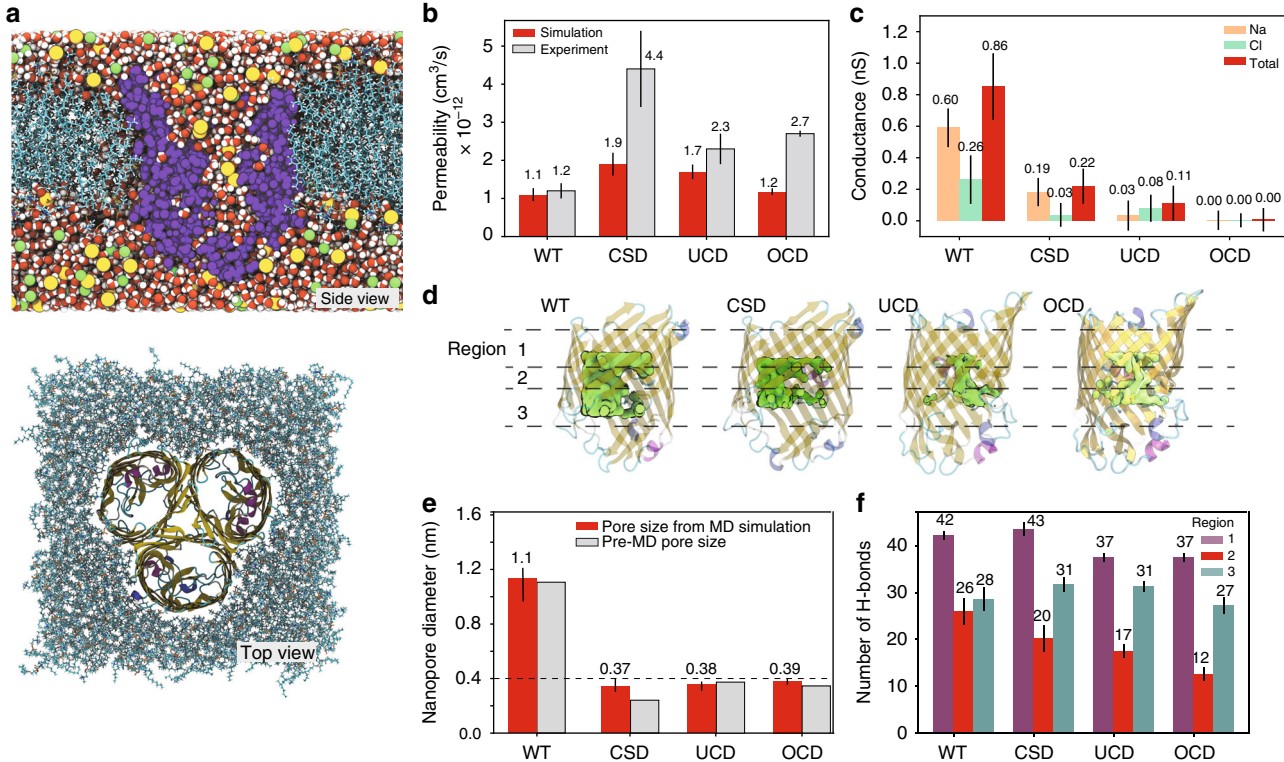

**Fig. 5** MD simulations of OmpF corroborate experimentally observed permeability and selectivity trends. **a** Typical simulation system. (Top) Cut-away view of the system revealing a transmembrane water passage through an OmpF monomer. The OmpF monomer is depicted in purple, the lipid-bilayer in cyan, water molecules as red and white spheres, and Na$^+$ and Cl$^-$ ions as orange and green spheres, receptively. (Bottom) Top-down view of the system. The OmpF trimer is drawn using a cartoon representation, the lipid-bilayer as cyan bonds; water and ions are not shown. **b** Simulated osmotic permeability (averaged over 12,500 frames) of OmpF variants (red) and the corresponding experimental values (gray). **c** Ionic conductance of OmpF trimers obtained from applied field simulations under a 500 mV transmembrane voltage and averaged over 10,417 frames. **d** Water occupancy of OmpF variants. The green volume depicts the average location of water molecules in each channel characterized as a 0.3 g/cm$^3$ isosurface of water oxygen density. For reference, each channel is shown using a semitransparent cartoon representation. **e** Major axis dimensions of the pores measured from PoreDesigner before MD (gray) and from the last 100 frames of MD (red). The error bars represent standard deviations. A 0.4 nm line represents the PoreDesigner design constraint of identifying pore designs smaller than 0.4 nm. **f** The average number of hydrogen bonds made between water and an OmpF monomer in each of the regions depicted in panel **d** and averaged over 14,583 frames

35 g/L NaCl). Similarly, for brackish water desalination this maximum is 0.034 nS (assuming a feed of 5 g/L NaCl), and 0.056 nS for low salinity wastewater (assuming a feed salinity of 2 g/L NaCl). All these conductance target values can be met by the derived UCD and OCD designs as their simulated conductance values are in the requisite range. However, it is important to note that the error bars on these calculations are quite wide implying that direct experimental validation is still needed.

The use of OmpF provides two distinct advantages over the use of AQPs for the envisioned applications in aqueous separations. First, AQPs are arguably overdesigned for water desalination as they remove protons along with other monovalent and divalent ions while OmpF can be designed to pass protons while removing other ions. The requirement to reject protons imposes the need to have hydrogen bonding between translocating water molecules and the pore wall. Recent studies have indicated that an ideal water-conducting pore could transport water more efficiently if all hydrogen bonding between waters and the central section of the pore are eliminated[47]. Second, the higher permeability of OmpF over AQPs can be advantageous for water purification and desalination in specific instances where space is at a premium or where energy savings can be substantial. Ultrapermeable membranes, such as those based on OmpF, with high salt rejection suited for RO can considerably minimize energy requirements (~45%) or plant infrastructure (pressure vessels, up to 65%) in

streams with modest salinity[9]; for instance brackish water desalination and recycling. Even though the energy advantage is marginal for high salinity seawater applications (15% less energy), there is a considerable plant size reduction (i.e., 44%[9,10]). Sub-nm pore size membranes for nanofiltration (NF) and ultrafiltration (UF) have diverse applications in water treatment[54], food production[55] and processing, and energy applications[4] which will also benefit from energy and capital cost reduction.

We demonstrated that PoreDesigner can precisely design sub-nm pore sizes within the stable beta barrel of a bacterial channel porin without jeopardizing the channel stability. PoreDesigner provides another powerful demonstration of de novo protein design on a class of proteins that so far have not been the subject of systematic protein design as have enzymes[56], antibodies[57], binding proteins[42] or protein interfaces. We experimentally validated our designs and show excellent solute selectivity for a range of solutes while maintaining high permeabilities. We subsequently expanded upon the range of designs obtained by PoreDesigner with pore sizes spanning the entire sub-nm (3–10 Å) spectrum in 1 Å bin sizes. PoreDesigner (Fig. 3) identified designs that spanned the entire pore size spectrum. OCD designs were seen to span the entire sub-nm spectrum whereas CSD and UCD designs were limited to the mid of the sub-nm spectrum. While these latter designs have not been experimentally tested yet and need to be further validated, successful computation-driven

designs for the 4 Å range suggest that the proposed framework lays the foundation of a paradigm for membrane-based sub-nm aqueous separations with applications ranging from desalination, vesicle-mediated drug delivery to separating solutes of biochemical importance with marginal difference in sizes. Moving beyond OmpF, we believe that PoreDesigner could be applied to tune pore size and geometry for any other porin system with a known structure.

## Methods

**PoreDesigner implementation.** PoreDesigner used results from molecular dynamics simulations (all-atom 10 ns with 2 fs timestep) of pressure-driven water transport through tetrameric AQP1-membrane assembly to isolate water wires that were then used to constrain the OmpF redesign process using an iterative design procedure where the interaction energy between the water wire and the mutated residues was maximized. By imposing a minimum percentage (50%) required of long side-chain residues in the redesigned pore, we safeguard against trivial designs involving all alanines or valines that would be too distant to interact with the water wire. We appended a design assessment step at each iteration by accepting only designs whose constriction diameter is less than 4 Å (Supplementary Methods for calculation). The accepted designs were classified as OCD, UCD, and CSD types dictated by their internal pore architecture. One design per type which showed lowest in silico pore area was used for experimental assessment of transport properties. The probable trend of water permeabilities was predicted by evaluating the inner and outer pore wall hydrophobicities. The respective hydrophobicity values (for the three top designs and WT) were computed by summing the transfer-free energy[58] ($\Delta G_{transfer}^{water \rightarrow ethanol}$) (from water to ethanol) of each one of the amino acid side chains that constitute the inner and outer pore wall. Lou et al.[33] define the inner and outer pore wall residues as those with side chains protruding into and out from the beta-barrel, respectively.

Further details, including a step-by-step procedure is provided in Supplementary Methods and Supplementary Figure 2 illustrates the cascade of steps used for pore profile analysis using a standalone PoreAnalyzer module of PoreDesigner.

**Molecular dynamics simulations.** All MD simulations were performed using the program NAMD, a 2 fs integration timestep, and 2–2–6 multiple time-stepping. Parameters for the POPC lipid-bilayer, OmpF protein, and ions were taken from the CHARMM36 parameter set with the CMAP corrections[59]. A TIP3P model was used for water[60]. All simulations employed a 10–12 Å cutoff for van der Waals and short-range electrostatic forces, the particle mesh Ewald method for long-range electrostatics[61] computed over a 1.1 Å grid and periodic boundary conditions. Equilibration simulations were performed in the NPT (constant number of particles N, pressure P, temperature T) ensemble using a Lowe–Andersen thermostat[62] and Nosé–Hoover Langevin piston pressure control set at 295 K and 1 atm, respectively. The simulations of ion conductance were carried out in the NVT (constant number of particles N, volume V, temperature T) ensemble. Visualization and analysis were performed using VMD[49]. Additional details are provided in Supplementary Methods.

**Experimental methods.** OmpF and the three selected mutant design proteins were produced by homologous expression of synthetic genes using pET23a(+) in an *E. coli* BL21(DE3) Omp8 Rosetta mutant (Genotype: F- ompT hsdSB(rB- mB-) gal dcm (DE3) pRARE (CamR) ompR ΔlamB ompF::Tn5 (KanR)) strain according to the pET cloning and expression system (Novagen). We obtained this strain from Dr. Roland Benz, Wisdom Professor of Department of Biotechnology, Jacobs University, in Bremen, Germany. Thereafter the porins were purified in their native trimeric state and stabilized in a detergent solution before reconstitution, and subsequently passed through an equilibrated anion exchange chromatography column (HiScreen DEAE HF), and a Superose 12 size exclusion column. Bradford assays were used to determine the protein concentration. Dry lipid films and a rehydration buffer were used to reconstitute mutant and WT OmpF which were extruded using a 200 nm track-etched membrane. Stopped-flow light scattering goniometer setup was used to assess hydraulic permeabilities. Finally, high concentrations of various solutes of different hydrodynamic radii were used in the mixing cell of the apparatus to determine solute rejection performance of the mutants in contrast to the WT OmpF.

**OmpF expression and cell culture.** Wild-type OmpF and OmpF mutant proteins were produced using the pET cloning and expression system (Novagen). Gene sequences for the WT and mutant OmpF protein designs were codon optimized, synthesized and cloned by GenScript USA (Piscataway, NJ). The synthesized genes were cloned into the *NdeI-XhoI* sites of the pET23a(+) expression vector and plasmids were maintained in *E. coli* TOP10. Purified plasmids were transformed into *E. coli* BL21(DE3) Omp8 Rosetta (ΔlamBompF::Tn5 ΔompAΔompC) mutant strain (gift from Professor Dr. Roland Benz). Four transformant strains were isolated: OmpF-WT, OmpF-OCD, OmpF-CSD, and OmpF-UCD. Each strain was

grown in Luria–Bertani (LB) with Cloramphenicol (20 mg/L) and Ampicillin (100 mg/L) at 37 °C (in batch culture). Once OD600 reached 0.5~0.7, Isopropyl beta-D-1-thiogalactopyranoside (IPTG) was introduced to the cell culture media at final IPTG concentration of 0.4 mM, then the incubation temperature was decreased to 16 °C for gene expression and protein production. After 12–16 h of cell growth, *E. coli* cells can be harvested and stored at −80 °C.

**OmpF purification.** OmpF mutants were purified[63] and 10 g frozen cells were first dissolved in 100 mL lysis buffer (20 mM Tris, 0.1 mg/mL DNase I, pH8.0) and lysed with an ultrasonic homogenizer. The mixture was spun down at 4000 × g for 20 min to separate unbroken cells. The lysed cells (in lysis buffer) were mixed with SDS at a final SDS concentration of 0.5% (wt/v) for 20 min at 4 °C and centrifuged at 200,000 × g for 60 min to harvest cell membrane pellets. Cell membrane pellets were suspended in 0.125% Octyl-POE, 20 mM Na₃PO₄, pH7.4 buffer (50 mL buffer/10 g original cells), and then cell membrane pellet mixture was incubated at 37 °C for 60 min. Then the membrane pellet mixture was centrifuged at 200,000 × g for 60 min in a Thermo Sorvall WX ultracentrifuge. The pellet was resuspended in 3% Octyl-POE, 20 mM Na₃PO₄, pH7.4 buffer (25 mL buffer/10 g original cells), and the mixture was incubated at 4 °C overnight. On the second day of the purification, the mixture was incubated at 37 °C for 1 h before ultracentrifugation (200,000 × g, 30 min) to spin down insolubilized cell membranes. The supernatant was collected after ultracentrifugation for chromatographic separation.

In the next step, an anion exchange chromatography column was used (HiScreen DEAE FF). This column was first equilibrated with 5 mM Na₃PO₄, 1% Octly-POE, 3 mM NaN₃, pH7.6 buffer. Then the supernatant previously collected was loaded into the column and eluted with 5 mM Na₃PO₄, 1% Octly-POE, 3 mM NaN₃, 30 mM EDTA, 100 mM NaCl, pH7.6 buffer. The elution peak fractions were combined and loaded onto Superose 12 size exclusion column (equilibrated with 10 mM Tris, 0.1 M NaCl, 1.2% Octyl glucoside) as a final purification step. The size exclusion peak fractions were collected for further use. Protein concentration was measured by Bradford assay.

**OmpF reconstitution into vesicles.** OmpF can be reconstituted into lipid vesicles[64] and allows passive diffusion of small molecules across the membrane. We characterized the influx of different molecular weight solutes through WT OmpF and its mutants under hypertonic conditions using stopped-flow light scattering measurements and compared their solute transport trends. All OmpF mutants were reconstituted into L-α-phosphatidylcholine (PC)/L-α-phosphatidylserine (PS) vesicles using a detergent destabilization method at LPR400[29]. Briefly, vesicles were created by rehydrating dry lipid films in rehydration buffer (20 mM HEPES, 100 mM NaCl, 0.02% (wt/v) NaN₃, pH7.4) and then extruded through 200 nm track-etched membranes. To these monodisperse vesicles, predetermined amounts of 10% (wt/v) Decyl Maltoside and rehydration buffer were introduced to "loosen" the vesicles to allow for efficient membrane reconstitution. OmpF protein was added to the detergent/vesicles mixture for reconstitution, and detergent in the mixture was removed by adding a predetermined amount of Bio-Beads™ SM-2 resin[65]. After detergent removal, the proteoliposomes were rapidly mixed with hypertonic solutions in the mixing cell of a stopped-flow setup, NaCl, glycine, glucose, sucrose, and polyethylene glycol 600 (PEG600) as osmolytes. All experiments were conducted with proteoliposomes with a polydispersity index of <0.2 leading to a signal to noise ratio of >50 in the stopped-flow curves that were used for permeability calculation. This is expected to provide high reliability in terms of the calculated parameters from this experiment[3].

The permeability values of vesicle membranes containing various mutant OmpF proteins were calculated from the light scattering intensity curves obtained from stopped-flow light scattering experiments conducted with completely excluding solute PEG600 as an osmotic agent for all mutants. By fitting the normalized light scattering intensity curve to a double exponential curve[19], we obtained a rate constant k (the larger constant from the double exponential fit). This rate constant was then used to calculate the osmotic water permeability ($P_f$) using equation 1 [19]:

$$P_f = \frac{k}{(S/V_o) \times \Delta\pi \times V_w},\qquad(1)$$

where $S$ is the initial surface area of OmpF-reconstituted vesicles, $V_o$ is the initial volume of OmpF-reconstituted vesicles, $\Delta\pi$ is the osmotic gradient across the lipid bilayers, and $V_w$ is the molar volume of water. Single-channel permeability was calculated by combining the net permeability from stopped-flow light scattering measurement with the number of proteins inserted per vesicle, determined from fluorescence correlation spectroscopy (FCS) experiments. This approach, pioneered by Pohl and co-workers[66], has been used to calculate the single-channel permeability of Aquaporin Z[29,46] and peptide-appended pillar[5]arene channels[67] successfully. OmpF proteins were first labeled using a pyrylium dye, which is shown to only have detectable fluorescence signal after conjugation with proteins[29]. We reconstituted these labeled OmpF proteins into vesicles and performed FCS to first determine the number of vesicles ($N_{ves}$) by fitting the auto correlation function obtained from FCS measurements of these fluorescent vesicles. We then added the membrane protein-compatible detergent, octyl glucoside (OG) to the same vesicle solutions (final OG concentration is 2.5%) to break down the vesicles into protein-detergent micelles. Thus, we could calculate the number of proteins ($N_{pro}$) using

the same method by conducting FCS measurements on these solubilized proteins assuming one protein trimer per micelle similar to what has been reported for aquaporins[46].

**Stopped-flow experiments**. OmpF (or OmpF mutants)-reconstituted vesicle solute rejection was estimated using experiments on a stopped-flow light scattering instrument (Supplementary Figure 3), based on a protocol similar to those employed for ion permeability studies and aquaporin permeability studies[46,68]. Vesicles were rapidly mixed with a high solute concentration osmotic agent (based on different tested solutes) in the mixing cell of this instrument. The high solute concentration osmotic agent included: 20 mM HEPES, 110 mM NaCl, 0.02% (wt/v) NaN$_3$, pH7.4 (for NaCl rejection measurement); 20 mM HEPES, 100 mM NaCl, 20 mM Glycine, 0.02% (wt/v) NaN$_3$, pH7.4 (for glycine rejection measurement); 20 mM HEPES, 100 mM NaCl, 20 mM glucose, 0.02% (wt/v) NaN$_3$, pH7.4 (for glucose rejection measurement); 20 mM HEPES, 100 mM NaCl, 20 mM sucrose, 0.02% (wt/v) NaN$_3$, pH7.4 (for sucrose rejection measurement); 20 mM HEPES, 100 mM NaCl, 15 mM PEG600, 0.02% (wt/v) NaN$_3$, pH7.4 (for PEG600 rejection measurement). Here, we used 15 mM PEG600 in PEG600 rejection measurement experiments to keep the same buffer osmolarity as other high solute concentration osmotic agents (measured in a freezing point osmometer).

For assessing solute rejection, proteoliposomes were subjected to a hyperosmotic shock, and time-dependent light scattering data collected upon mixing of the osmolyte and proteoliposomes. The resulting light scattering profile was used to determine solute exclusion as well as water permeability of incorporated proteins. As shown in Fig. 4, for both the cases where solute is completely excluded by the channel (solute exclusion model) and when there is some solute leakage through the channel (solute permeable model), during the first stage of the mixing process water flows outward from the vesicles leading to vesicle shrinkage, due to the high osmolarity outside the vesicles. This shrinkage leads to an increase of light scattering intensity measured at 90° to the incident light due to constructive interference of scattered light. This is because vesicles with a size comparable to the wavelength of light stop acting like point particles and show an increasing trend in scattering intensity with decreasing volume[69] at the scattering angle used for measurements (90°). During the second stage of the mixing process (Fig. 4), the light scattering intensity trend changes based on whether the solute can or cannot not diffuse through the porin[68]. When the solute molecular size is larger than the porin pore size (Fig. 4, solute exclusion model), the water continues flowing outward from the vesicles to reach the equilibrium state of the osmotic pressure dictated by the solute concentration, and the light scattering intensity levels off as measurement time increases. However, if the solute size is smaller than the porin pore size (Fig. 4, solute permeable model), in the second stage solutes diffuse through porins and led to a corresponding influx of water into vesicles, which is observed as a decrease in light scattering intensity[68]. Based on the observation of light scattering intensity change, we could estimate the solute rejection trends of porins.

The detailed calculations from an example case for determining maximum OmpF conductances for removal of salt from various feed streams have been reported in Supplementary Methods and Supplementary Table 2.

## Data availability

Data supporting the findings of this manuscript are available from the corresponding authors upon reasonable request. PoreDesigner can be downloaded from the research github repository (https://github.com/maranasgroup) and webpage (http://www.maranasgroup.com/software.htm) of Costas D. Maranas. A separate PoreAnalyzer module for analyzing pore profile of any porin (alpha helical or beta barrel) will also be available from the webpage.

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

## Acknowledgements

The authors acknowledge financial support from the National Science Foundation CAREER grant (CBET-1552571) to M.K. for this work. Support was also provided through CBET-1705278, DMR - 1709522, and NSF MCB-1359634 for various aspects of this work. *E. coli* BL21(DE3) Omp8 Rosetta (ΔlamBompF::Tn5 ΔompAΔompC) mutant strain was a gift from Professor Dr. Roland Benz, Jacobs University, Bremen, Germany. The Fluorescence Correlation Spectroscopy experiments were performed in Professor Peter Butler's Lab.

## Author contributions

M.K., C.D.M., R.C. and T.R. conceived the study. Protein redesign simulations were performed by R.C. and experiments were performed by T.R. and J.P. M.G. helped in designing and setting up the simulation and pore size calculation for all the designs. M.S. and K.D. performed the MD simulations and M.S. and A.A. helped to analyze the MD simulation results. C.B. and J.H.G. assisted in cloning for mutants. All authors contributed to writing and editing the manuscript.

## Additional information

**Competing interests:** The authors declare no competing interests.

