## [Peer Review File · Nature Communications]

Reviewers' comments:

Reviewer #1 (Remarks to the Author):

The paper explores a rationally design a channel protein structure that will have a desired permeability and selectivity. The problem of engineering novel selective channels and controlling their properties has been attracting much attention over the last decade, even though it remains a purely scientific subject despite years of basic research. However, here the authors come up with a fresh approach, which they also successfully carried out. They managed to modify a naturally occurring poorly-selective channel protein OmpF to impart it with desired pore structure and make it more size-selective. The modifications were designed to reduce the pore size and change its hydrogen-bonding characteristics by adding appropriate bulky motifs to the protein sequence, verify the structure by molecular simulation, and experimentally confirm the permeation characteristics in stop flow experiments using protein-laden lipid vesicles.

I have to admit I doubt that protein channels can ever make it to separation or desalination technologies, since their stability and lifetime is too small, but that is my personal opinion. Still I find the idea of improving Nature in a rational way exiting, especially, sinnce the authors eventually manage to beat the Nature, i.e., native aquaporins designed to do essentially the same job. I then do think that this work is an important contribution to understanding the physics of separations, especially, when it supplies evidence that elimination of hydrogen-bonding in a narrow hydrophobic pore may enhance water permeation.

The authors also demonstrate selectivity, but I wonder if the "negligible" ion permeability was small enough to define the channels as selective. How small would be negligible? It could be worth to define what would be considered sufficient (e.g., in comparison to AQPs or existing membranes) and see how close the new channels come to that.

I have a few technical comments that may help improve the clarity of the manuscript and provide useful data for the future.

- Inner/outer pore wall – it is not very clear how these were distinguished, the SI file gives some vague idea, but it should state in a way clear not only for protein experts. Why would outer wall hydrophobicity contribute to water permeation, is water supposed to interact with it as well? (The answer might not be necessary, if it was clear what the outer wall is.)
- WT – (wild type?) –make sure these and other abbreviations are defined
- Important experimental details and numbers:
 - o Did the vesicles initially contain pure water before mixing with osmolites?
 - o Why the largest rate constant of the double-exponential fit (meaning short-time part of the curve?) was the relevant one? Was water permeation the only mechanism responsible for initial rapid shrinking?
 - o I suggest to explicitly state the average size of the vesicles in stop-flow experiments and specify, even as a range, how many proteins on average a vesicle contains (how certain was that?). These numbers may be important as future benchmarks.
- I understand that 0.3 nm/20 degrees criterion for assuming H-bond formation was necessary, since simulation were not QM. Is it well justified? Any ref that confirms this criterion versus, say, DFT simulations?
- Line 429 "dense solution, diffusion-based" – it takes a while to realize it must be "dense solution-diffusion-based". Overall, the punctuation in the manuscript is very untidy, which makes it harder to read.

Reviewer #2 (Remarks to the Author):

Summary of the Work:

The goal of this study was to design protein pores that are selective for different solutes. To accomplish this goal, the authors developed a computational workflow called PoreDesigner which

can redesign a beta-barrel pore to have various sizes. PoreDesigner was applied to design pores between 3-10 Å using OmpF as a scaffold. Then, pores between 3-4 Å were experimentally validated using stopped-flow light scattering. The study is novel because the computational design process yielded one scaffold with intended solute rejection behavior. This work is of broad interest to the membrane protein community, the protein design community, and others. However, the authors need to elaborate on the limitations of PoreDesigner because other pore sizes were not experimentally validated. Specific requested points of revision are below.

Major Revisions:

- (1) Title: "Dialing in" is an idiom that some readers may not recognize. A possible alternative is "PoreDesigner: Tuning solute selectivity in a robust and highly permeable outer membrane pore"
- (2) Introduction: Selectivity and permeability are key concepts in this manuscript. The introduction would benefit from 1-2 sentences quantitatively defining selectivity and permeability in context of membrane transport.
- (3) Introduction: Why did the authors choose OmpF compared to other outer membrane protein beta barrels? Are there literature values for the selectivity and permeability of WT OmpF?
- (4) Introduction: The introduction would be improved by describing other examples of membrane protein design. This would frame the novelty and difficulty of this study. Are there any other proteins that would be suitable scaffolds for designed permeable membranes? One example reference: John NH, Wang T, Bhate MP, Acharya R, Wu Y, Grabe M, Hong M, Grigoryan G, DeGrado R (2014) "De novo design of a transmembrane Zn²⁺-transporting four-helix bundle" *Science* 346 (6216): 1520-1524.
- (5) Introduction: The paragraph beginning on line 94 can be shortened to include less experimental details. Many of these (e.g. use of IPro, mixed-integer linear optimization) can be moved to the results. This may help make room for other additions later.
- (6) Results: The authors computationally design OmpF pore sizes between 3-10 Å. However, they only experimentally validate the smallest pores (3-4 Å). As a result, the authors cannot conclude that PoreDesigner can design the full range of pore sizes. As a computational scientist, I'm not familiar with the difficulty of the solute rejection experiments. If possible, the authors may choose to validate 1-2 designs for each pore size (e.g. 4-5 Å, 5-6 Å). Alternatively, the authors should rephrase these conclusions to indicate PoreDesigner can design small pores and include the additional solute rejection experiments as future work.
- (7) Results: How did the authors estimate the pore dimensions (i.e. where does 7x11 Å come from)? Is this a measurement of the minimum bounding ellipse excluding the side chain atoms?
- (8) Results: Is the PoreDesigner code a single package or a combination of multiple packages? If multiple packages, is there a single script or executable to unify the steps?
- (9) Results: Is the PoreDesigner code readily available? Would it be easy for an outside reader to reproduce the method? If not, can the code be released to the public upon publication?
- (10) Results: Why did the authors choose to only test the smallest pore size? How did they know this pore size is appropriate for rejecting the desired solutes?
- (11) Results: Why did the authors choose a water-to-ethanol transfer free energy scale to estimate hydrophobicity? Ethanol has a dielectric constant of ~24 and is thus not a great model of a very nonpolar environment such as the protein core (dielectric constant between 2-10) and a membrane (dielectric constant of 2).
- (12) Results (Line 177): How did the authors arrive at 50% as a reasonable limit for the number of alanines and valines? Did the authors test various values or is there precedence in the protein design literature?
- (13) Results (Line 227-229): How are clashes related to hydrophobicity? I'm having difficulty following the logic of this sentence.
- (14) Results: Why were the OmpF proteins arranged as trimers in the MD simulation? Is this what is observed in wild-type?
- (15) Discussion: Do the authors think their pore design strategy would extend well to other beta-barrel scaffolds? What about designed alpha-helical bundles such as this recent work from David Baker's lab? (Reference: Lu P, Min D, DiMaio F, Wei KY, Vahey MD, Boyken SE, Chen Z, Fallas JA,

Ueda G, Sheffler W, Mulligan VK, Xu W, Bowie JU, Baker D (2018) "Accurate computational design of multipass transmembrane proteins" Science 359 (6379) pp 1024-1046.)

(16) Discussion: What do you expect will happen if you perform the solute rejection experiments on OmpF with other designed pore sizes?

Minor Revisions:

(1) Line 116: Remove the statement in parentheses "(with molecular retention of solutes with molecular weights of 600 Da or larger)"

(2) Line 120: Why is there a \sim symbol for $8.97 \times 10^{-14} \text{ cm}^3/\text{s}$? This is confusing in the context of other quantities that include an uncertainty.

(3) Line 286: "(or glycine, glucose, and sucrose)" Parentheses are confusing about meaning.

(4) Typos:

- Line 173: "Hydrogen atoms and atoms from residues away"

- Line 199: "We identified were 31 UCD and seven CSD designs"

- Line 215: "Adding up free energy of transfer" (should be free energies)

Reviewer #3

In this work, Chowdhury *et al.* develop and apply a unique computational approach to redesign a bacterial porin (OmpF) for fast and selective water transport. Using computationally-directed mutagenesis, the authors design three distinct classes of OmpF mutants that achieve superior size-based solute exclusion relative to the wild type channel. Furthermore, the fastest water flux achieved by one of the OmpF variants exceeds the AQP1 water flux by ~ an order of magnitude, making these variants potentially suited for building stable and efficient biomimetic membranes for challenging water purification applications.

Major comments

1. **Line 122** – based on single channel permeability measurements, it would appear that the fold enhancement is more in the one order of magnitude range, not two.
2. **Line 206** – which OCD design is being referred to here? TFPhe or TFTrp or some other Trp/Phe pore ordering? This information would be useful. Additionally, it will help to clarify that the pore is elliptical and the dimensions refer to the major & minor axes.
3. **Lines 220-233** – I am having some trouble understanding this section (perhaps I am missing some thing). I expect the TFTrp OCD design to have the highest hydrophobicity, which is consistent with what the authors claim in line 231-232. However, line 221 asserts that CSD has greater hydrophobicity compared to OCD. Why would this be the case? With 25 Phe(s) or 25 Trp(s) or 25 Trp/Phe combinations, shouldn't OCD hydrophobicity surpass that of CSD? Also, lines 232-233 seem to attribute increased hydrophobicity in OCD to steric clashes between adjacent Trp(s) or Phe(s). I understand that steric clashes can cause lumen widening – but how do they contribute to increasing hydrophobicity?
4. **Lines 256-258** – The UCD design has a smaller pore than the CSD design by ~ 3%. However, from Figure 3b, the smallest pore size bin does not include any UCD design (only a CSD). It seems a bit difficult to reconcile this distribution with the fact that UCD makes provision for the narrowest pore. Could the authors shed some light on this?
5. **Line 262** – To what extent are the stopped flow measurements affected by polydispersity in the reconstituted vesicles? Although Prof. Kumar is an expert in this area, some discussion on this would benefit readers who are new to this experimental technique.
6. **Lines 282-284** – Is there a typing error? If the solute size is larger than the pore size, I would expect the solute to be rejected and not subject to an “inflow of water and solutes into vesicles” as claimed in the manuscript. Please correct/clarify.
7. **Figure 4c** – the stopped flow data for glycine & glucose appear noisier than corresponding data for sucrose and PEG600, although the trends are still pretty clear. I recommend that the authors provide some explanation to account for the different variabilities in these data sets. Could this be a result of different PDI(s) of various batches of reconstituted vesicles used in the experiments?

8. **Lines 302-303** – the stopped flow experiments are capped at the 5 second time point in Figs. 4c & d. Have the authors looked at longer times? Is it possible that tight mutants such as UCD allow solute leakage over time? If true, exclusion benefits of UCD would only be useful over short time-scales. I recommend that the authors address this potential issue by including longer time-scale experiments for NaCl exclusion by UCD. Additionally, it would be useful if the authors could compare their stopped flow time scales with typical times that might be employed in a realistic (*i.e.*, pilot scale) membrane desalination scenario.
9. **Line 358** – Is the 1 protein trimer/micelle assumption valid for OmpF, which has a different size and multimerization propensity compared to AQP?
10. **General comment** – While the work is truly exciting, it could be strengthened by including a deeper analyses of the key benefits of these engineered OmpF variants relative to AQP1 or RsAqpZ, perhaps in the context of membrane purifications. This would be in addition to the expanded solute selectivity benefit that the authors talk about in the Discussion section. For instance, is faster flux really critical for membrane applications? What kind of techno-economic (and other) benefits will a 40-fold faster channel confer relative to current aquaporin-based approaches? Furthermore, in the introduction, the authors allude to poor stability of AQP1 as being a limitation. Can they provide some experimental evidence of enhanced stability of the designed OmpF mutants relative to AQP1 for long term or repeated use? I am afraid that the hydrophobic residues in the channel pore could lead to an increased propensity for the pore to “collapse”? Have the authors performed experiments to exclude this possibility?

Minor Comments

1. **Line 74** – please include a relevant reference
2. **Line 77** – Could the authors furnish a reference that describes AQP1 stability as an issue that complicates membrane insertion or long-term use? If ref. 35 happens to be the relevant reference, this is not clear from the bibliography.
3. **Line 141** – using “molecular dynamics simulations”
4. **Line 180** – “redesigned” what?
5. **Figure 2, caption** – I think the authors meant to say “UCD designs intersperse smaller side chain hydrophobic amino acids”. Is that correct?

Reviewer’s decision

I recommend that the manuscript be accepted for publication in *Nature Communications* after the authors have addressed the concerns raised above.

Reviewer #1 (Remarks to the Author):

The paper explores a rationally design a channel protein structure that will have a desired permeability and selectivity. The problem of engineering novel selective channels and controlling their properties has been attracting much attention over the last decade, even though it remains a purely scientific subject despite years of basic research. However, here the authors come up with a fresh approach, which they also successfully carried out. They managed to modify a naturally occurring poorly-selective channel protein OmpF to impart it with desired pore structure and make it more size-selective. The modifications were designed to reduce the pore size and change its hydrogen-bonding characteristics by adding appropriate bulky motifs to the protein sequence, verify the structure by molecular simulation, and experimentally confirm the permeation characteristics in stop flow experiments using protein-laden lipid vesicles.

I have to admit I doubt that protein channels can ever make it to separation or desalination technologies, since their stability and lifetime is too small, but that is my personal opinion. Still I find the idea of improving Nature in a rational way exiting, especially, since the authors eventually manage to beat the Nature, i.e., native aquaporins designed to do essentially the same job. I then do think that this work is an important contribution to understanding the physics of separations, especially, when it supplies evidence that elimination of hydrogen-bonding in a narrow hydrophobic pore may enhance water permeation.

The authors also demonstrate selectivity, but I wonder if the “negligible” ion permeability was small enough to define the channels as selective. How small would be negligible? It could be worth to define what would be considered sufficient (e.g., in comparison to AQPs or existing membranes) and see how close the new channels come to that.

We thank the reviewer for their recognition of the novelty of this work. I am not sure if the reviewer is aware but there is a commercial aquaporin based membrane on the market and has been tested for several applications. See for example (<https://aquaporin.dk/products/ro/>). But the reviewer’s comment is well taken, how long these membranes will last and what the final verdict of the market be is still unclear. Our work was geared towards understanding whether the biophysics of transport in narrow channels can be further elucidated by using “by design” variations of angstrom scale channels.

With regard to the reviewer’s question on what could be considered negligible ion permeability, we conducted some simple theoretical analyses of what negligible ion permeability would be, so that membranes based on channels could match the performance of seawater RO membranes, brackish water RO membranes and wastewater recycling membranes. We estimated that the maximum total conductance values to be 0.018 nS for seawater desalination (assuming a feed of 35 g/L NaCl), 0.034 nS for brackish water desalination (assuming a feed of 5 g/L NaCl), and 0.056 nS for low salinity wastewater (assuming a feed salinity of 2g/L NaCl). All these targets can in principle be met by our UCD and OCD designs as their calculated conductance is within these ranges (note however that the error bars are large and further experimental validation would be required). An example calculation is included in the Supporting Information and the following language inserted in the main manuscript text

“A simple theoretical analysis of what ion conductance targets would be achievable for membranes based on OmpF channels is described in the Supporting Information. The analysis reveals that for seawater desalination, the estimated maximum total conductance would be 0.018 nS for seawater desalination (assuming a feed of 35 g/L NaCl). Similarly for brackish water desalination this maximum would be 0.034 nS (assuming a feed of 5 g/L NaCl), and 0.056 nS for low salinity wastewater (assuming a feed salinity of 2g/L NaCl). All these targets can be met by the UCD and OCD designs as their estimated conductance is within the range simulated for these mutants (note however that the error bars on these calculations are large and further experimental validation would be required).”

I have a few technical comments that may help improve the clarity of the manuscript and provide useful data for the future. - Inner/outer pore wall – it is not very clear how these were distinguished, the SI file gives some vague idea, but it should state in a way clear not only for protein experts. Why would outer wall hydrophobicity contribute to water permeation, is water supposed to interact with it as well? (The answer might not be necessary, if it was clear what the outer wall is.)

Any OmpF residue with its side chain facing the membrane is referred to as an outer-pore wall residue whereas the ones with side chains facing the pore lumen as inner-pore wall residue. Residues were classified accordingly by analyzing whether the first atom of the side chain of any residue (C β atom for others and H α for glycines) is closer to the central pore axis than the C α atom of the residue’s backbone. If yes, then we classified a residue to be inner-pore wall or lumen-facing, otherwise it was referred to as outer-pore wall or membrane facing. Note that only inner pore wall residues contribute to inner wall hydrophobicity which affects hydraulic permeability through the redesigned porin.

Lou *et al.*¹ (Journ. Biol. Chem, 1996) have performed extensive structural and functional characterization of OmpF and its various mutants, where they define the inner pore wall residues as “residues that protrude into the barrel” and outer pore wall residues as “residues that protrude from the barrel”. They indicate that, the transport properties of the mutant porins could be altered by altering the pore constriction “residues that protrude into the barrel”. They explore ion selectivities of five such OmpF mutants. We have inserted this reference and some text in the revised manuscript as a justification for correlating inner pore wall hydrophobicity scores to predicted water permeation rates, across these porins.

- WT – (wild type?) –make sure these and other abbreviations are defined

All abbreviations including WT are now defined appropriately (line 86). Wild type (WT) is defined in the Abstract line 28 of the revised manuscript.

- Important experimental details and numbers:

o Did the vesicles initially contain pure water before mixing with osmolites?

The vesicles initially contained both water and rehydration solutes as they were rehydrated with buffer (20mM HEPES, 100mM NaCl, 0.02% (wt/v) NaN₃, pH7.4).

o Why the largest rate constant of the double-exponential fit (meaning short-time part of the curve?) was the relevant one? Was water permeation the only mechanism responsible for initial rapid shrinking?

Based on extensive previous studies that have applied the stopped flow light scattering technique to measure the permeability of aquaporin incorporated liposomes^{2,3}, permeability was calculated based on the assumption that the osmotic pressure difference across the bilayer is constant, which is valid only at the beginning of measurement. In general, the first 0.2-0.5s of these stopped flow light scattering curves were fitted for this study (for a total measurement time of 5s), depending on the kinetics of the shrinkage. When applying a double-exponential fit to this part of the curve at short measurement times, the rapid increase of the light scattering was mainly contributed by the large rate constant part of the equation. Thus, in both the previous aquaporin studies and our OmpF study in this paper, the large rate constant of the double exponential fit was applied to calculate water permeability. Under our current stopped flow measurement, water permeation is the only mechanism responsible for initial rapid shrinking

o I suggest to explicitly state the average size of the vesicles in stop-flow experiments and specify, even as a range, how many proteins on average a vesicle contains (how certain was that?). These numbers may be important as future benchmarks

The average diameter of the vesicles in the stopped flow experiments was 175 (±13) nm. Based on our FCS measurements, the number of proteins per vesicle for UCD is 38 (±6), the number of proteins per vesicle for CSD is 16 (±7), the number of proteins per vesicle for OCD is 18 (±2) and the number of proteins per vesicle for WT is 19 (±7). These data are provided in the revised supporting information (Figure S3). Note that in response to another comment, we conducted a number of new experiments to determine if signal to noise can be improved in our experiments, these data are included in the new version of the manuscript (but do not change any conclusions or interpretations of the data).

I understand that 0.3 nm/20 degrees criterion for assuming H-bond formation was necessary, since simulation were not QM. Is it well justified? Any ref that confirms this criterion versus, say, DFT simulations?

Ireta *et al.*⁴ (JPCA 2004) and Durrant *et al.*⁵ (JMGM, 2011) report a criterion of the distance between the acceptor (A) and donor (D) molecule below 0.35 nm and the angle between A-D-H where H denotes the hydrogen atom ranging from 0 to 35 degrees. The 0.3 nm D-A distance and 20 degrees AsD-H angle criterion cutoff measures moderate hydrogen bond energies greater than ~4 kcal/mol as reported by Ireta *et al.* in DFT simulations. The following citations are now included in the revised manuscript.

-Line 429 “dense solution, diffusion-based” – it takes a while to realize it must be “dense solution-diffusion-based”. Overall, the punctuation in the manuscript is very untidy, which makes it harder to read.

We have made these corrections and worked on punctuation in the manuscript.

Reviewer #2 (Remarks to the Author):

Summary of the Work:

The goal of this study was to design protein pores that are selective for different solutes. To accomplish this goal, the authors developed a computational workflow called PoreDesigner which can redesign a beta-barrel pore to have various sizes. PoreDesigner was applied to design pores between 3-10Å using OmpF as a scaffold. Then, pores between 3-4 Å were experimentally validated using stopped-flow light scattering. The study is novel because the computational design process yielded one scaffold with intended solute rejection behavior. This work is of broad interest to the membrane protein community, the protein design community, and others. However, the authors need to elaborate on the limitations of PoreDesigner because other pore sizes were not experimentally validated. Specific requested points of revision are below.

We have added a caveat in the discussion section indicating that only a selected range of pore sizes was tested. We do feel however sub 4 Angstrom pores were the most stringent we could have conducted since it is the most selective aqueous separation possible, particularly when the large native pore size of this protein of 11 Angstrom is considered.

Major Revisions:

(1) Title: “Dialing in” is an idiom that some readers may not recognize. A possible alternative is “PoreDesigner: Tuning solute selectivity in a robust and highly permeable outer membrane pore”

We agree with the reviewer’s suggestions and changed the paper title accordingly in the revised manuscript.

(2) Introduction: Selectivity and permeability are key concepts in this manuscript. The introduction would benefit from 1-2 sentences quantitatively defining selectivity and permeability in context of membrane transport.

We agree with the reviewer’s suggestion. We include the following definitions in the revised manuscript:

“Permeability and selectivity are key concepts for membrane separations. Membrane permeability is defined as the membrane flux (volume of liquid or gas that passes per unit membrane area per unit time⁶) per unit driving force. For biological membranes, a non-dimensional driving force is used (osmotic gradient times the molar volume of water) – this is the permeability reported in this work⁷. Membrane selectivity is commonly defined as the ratio between the permeability of two solutes, two solvents or between a solute and a solvent.”

(3) Introduction: Why did the authors choose OmpF compared to other outer membrane protein beta barrels? Are there literature values for the selectivity and permeability of WT OmpF?

This is an excellent question. As discussed in the manuscript, OmpF is a protein with high stability (mechanical and chemical). It also has high mutation stability as evidenced by the many studies that have studied various mutations of this protein. It also has the ideal pore size of ~1nm that can be a test case for sub nm pore design. Other Outer membrane channels such as FhuA have also been extensively mutated but the native pores size is 2 nm, making sub nm pore design much more challenging. We agree that channels such as OmpA, OmpG and OmpX could also be excellent targets for redesign, we simply chose OmpF because of the large amount of information on solute permeability available on this channel. Note however that a careful analysis of the single channel water permeability has not been conducted before.

(4) Introduction: The introduction would be improved by describing other examples of membrane protein design. This would frame the novelty and difficulty of this study. Are there any other proteins that would be suitable scaffolds for designed permeable membranes? One example reference: John NH, Wang T, Bhate MP, Acharya R, Wu Y, Grabe M, Hong M, Grigoryan G, DeGrado R (2014) “De novo design of a transmembrane Zn²⁺-transporting four-helix bundle” Science 346 (6216): 1520-1524.

Membrane protein redesign studies have been performed and reported on gated ion channels (Bocquet *et al.*⁸, Hibbs and Gouaux⁹, Hilf and Dutzler¹⁰, Miyazawa *et al.*¹¹). In a recent study, Liu *et al.*¹² engineered a ferrichrome outer membrane iron transporter (FhuA) to attain pore sizes of 1.6nm to 2.7nm and explored its transport properties.. Fairman *et al.*¹³ provides a list of known beta-barrel proteins with resolved crystal structures (along with their corresponding PDB ids). Depending upon ease of experimental validation, any of these proteins could in principle be redesigned using

PoreDesigner. This text has been appended (Introduction line 73 and Discussion line 495) in the revised manuscript and the suggested John *et al.* (2014) paper has been appropriately cited in the introduction.

(5) Introduction: The paragraph beginning on line 94 can be shortened to include less experimental details. Many of these (e.g. use of IPRO, mixed-integer linear optimization) can be moved to the results. This may help make room for other additions later.

As per our reviewer's suggestion we moved most of IPRO details to the Results section.

(6) Results: The authors computationally design OmpF pore sizes between 3-10 Å. However, they only experimentally validate the smallest pores (3-4 Å). As a result, the authors cannot conclude that PoreDesigner can design the full range of pore sizes. As a computational scientist, I'm not familiar with the difficulty of the solute rejection experiments. If possible, the authors may choose to validate 1-2 designs for each pore size (e.g. 4-5 Å, 5-6 Å). Alternatively, the authors should rephrase these conclusions to indicate PoreDesigner can design small pores and include the additional solute rejection experiments as future work.

The reviewer is correct in stating that Poredesigner was only validated for pore sizes between 3-10 Å. Validating Poredesigner for the complete range is the subject of current investigations and the topic of a follow up article. The revised manuscript discussion section is updated to better reflect this.

(7) Results: How did the authors estimate the pore dimensions (i.e. where does 7×11 Å come from)? Is this a measurement of the minimum bounding ellipse excluding the side chain atoms?

We performed a non-linear regression on the side chain hetero atoms using the general equation of an ellipse and identified the maximum bounding ellipse that excludes all these atoms.

(8) Results: Is the PoreDesigner code a single package or a combination of multiple packages? If multiple packages, is there a single script or executable to unify the steps?

Yes, PoreDesigner is a single package built inside the IPRO Suite of programs¹⁴. It can be downloaded and used using a Creative Common license. It has software dependencies on CHARMM and CPLEX/GAMS.

(9) Results: Is the PoreDesigner code readily available? Would it be easy for an outside reader to reproduce the method? If not, can the code be released to the public upon publication?

Yes, PoreDesigner would be available for download from <http://www.maranasgroup.com/software.htm> upon publication. However, a user requires CHARMM and CPLEX/GAMS licenses for running PoreDesigner. Also, the user would need to change the default paths to the CHARMM and GAMS executables before running.

(10) Results: Why did the authors choose to only test the smallest pore size? How did they know this pore size is appropriate for rejecting the desired solutes?

The smallest pore sizes were selected as they were perhaps the most challenging demonstration for this channel (the WT size was ~ 1nm pore size and molecular weight cut-off (MWCO) of 600 Da) as well as the high impact of the possible separations that could be conducted with this small pore size. Small pore size OmpF mutants offer promise as biomimetic membranes with sub-nanometer pore size that can be used to separate small molecules with molecular weight below 1000Da, a challenging task for most current membranes. For example, pharmaceutical applications could include (i) the concentration of ampicillin (349Da), and (ii) the separation of lower molecular weight genotoxic impurities (55-225Da) from higher molecular weight active pharmaceutical ingredients (170-840Da)¹⁵.

In order to know the solute rejection of designed OmpF mutants, we covered a range of solutes with different molecular weights, from 58.5 Da to 600 Da and performed solute rejection experiments. Since OmpF mutants we selected in this paper were designed to have pore size in between Aquaporin and wild type OmpF to demonstrate that we met our design goals, we selected NaCl (58.5Da) as our smallest test solute since Aquaporin is known to reject NaCl, while we selected PEG600 (600Da) as our largest test solute since wild type OmpF is known to reject molecules with 600Da or higher molecular weight.

(11) Results: Why did the authors choose a water-to-ethanol transfer free energy scale to estimate hydrophobicity? Ethanol has a dielectric constant of ~24 and is thus not a great model of a very nonpolar environment such as the protein core (dielectric constant between 2-10) and a membrane (dielectric constant of 2).

The Kyte-Doolittle method for discerning protein structure hydrophobicities using ethanol-water free energies has been used extensively in literature and has even been used as a standard to assess the performance of novel hydrophobicity scales by Perunov *et al.*¹⁶. Furthermore, Kister *et al.*¹⁷ has reported that the accuracy of the KD scale in estimating protein hydrophobicities is considerably reliable. This is clarified in the revised Supplementary Information of the manuscript.

(12) Results (Line 177): How did the authors arrive at 50% as a reasonable limit for the number of alanines and valines? Did the authors test various values or is there precedence in the protein design literature?

While designing small pore sizes using PoreDesigner, we first aimed at getting the narrowest pore size possible (using the OCD-TFTrp and OCD-TFPhe designs). Upon discovering that steric clashes prevent these designs from having the narrowest sizes, we allowed the pore constriction residues to be altered to other hydrophobic residues. However, allowing unconstrained use of alanines and valines would yield designs with larger pore sizes that will be ultimately weaned out at the pore size-checking step. This would necessitate more iterations of PoreDesigner to ultimately choose long-side chain residues and accepted designs. Thus, this 50% constraint is added to accelerate convergence. Some testing confirmed that 50% is a reasonable compromise. Having said that, the code is flexible to allow the users to specify any percentage value. This is explained in the revised PoreDesigner methods section of the Supplementary information.

(13) Results (Line 227-229): How are clashes related to hydrophobicity? I'm having difficulty following the logic of this sentence.

OCD-TFTrp designs have most of the 25-tryptophan side-chains sterically clashing with one another. This causes some of the clashing Trp side chains to face away from the pore lumen and become membrane-facing. These residues constitute the OmpF outer pore wall and contribute to the outer pore wall hydrophobicity. As a result, even though 25 Trps are present at the pore constriction region, only nine out of 25 of them face the pore lumen. Thus, the inner pore wall (which is in contact with the permeating water wire) becomes less hydrophobic than expected.

(14) Results: Why were the OmpF proteins arranged as trimers in the MD simulation? Is this what is observed in wild-type?

OmpF is indeed a trimeric protein. We have added this in line 96 of the Introduction of the updated manuscript which reads: "In particular, we worked with the trimeric *Escherichia coli* protein outer membrane porin type F (OmpF, wild type pore size of ~ 11Å) to attain desired pore sizes that could enable precise molecular separations".

(15) Discussion: Do the authors think their pore design strategy would extend well to other beta-barrel scaffolds? What about designed alpha-helical bundles such as this recent work from David Baker's lab? (Reference: Lu P, Min D, DiMaio F, Wei KY, Vahey MD, Boyken SE, Chen Z, Fallas JA, Ueda G, Sheffler W, Mulligan VK, Xu W, Bowie JU, Baker D (2018) "Accurate computational design of multipass transmembrane proteins" *Science* 359 (6379) pp 1024=1046.)

We believe that in principle we can use PoreDesigner to redesign any beta-barrel scaffold and likely alpha-helical proteins. Our method does not distinguish between alpha helix bundles or beta-barrels; it only requires (1) a list of residue positions that could be altered and, (2) a list of allowed amino acids in each of these positions. However, since we have not tested PoreDesigner for other systems we have refrained from such assertions.

(16) Discussion: What do you expect will happen if you perform the solute rejection experiments on OmpF with other designed pore sizes?

Based on our understanding of the solute rejection experiments, if the OmpF mutants have pore sizes in between OmpF_OCD and OmpF_WT, the stopped flow light scattering intensity curve will decrease during the "second stage" when exposing mutant reconstituted vesicles to sucrose containing hypertonic solutions. And the stopped flow light scattering intensity curve would level off when exposing mutant reconstituted vesicles to PEG600 containing hypertonic

solutions at the second stage. However, we may also need to include the solutes that have molecular weight in between 342Da to 600Da in the solute rejection experiments to get more solute rejection information for the newly designed pore size range.

Minor Revisions:

(1) Line 116: Remove the statement in parentheses “(with molecular retention of solutes with molecular weights of 600 Da or larger)”

Based on the reviewer’s suggestion, we rewrote the sentence as follows:

“We were able to redesign the 7×11Å OmpF WT pore to obtain an array of designs with varying pore size profiles sampling pore sizes across the 3 – 10 Å range and experimentally tested a subset of these designs in the 3-4 Å range critical for most challenging separations.”

(2) Line 120: Why is there a ~ symbol for $8.97 \times 10^{-14} \text{ cm}^3/\text{s}$? This is confusing in the context of other quantities that include an uncertainty.

This is the reported value of AQP1 single channel permeability. We cited this value from reference “Aquaporin water channels: atomic structure molecular dynamics meet clinical medicine”¹⁸, in this paper, the author claimed AQP1 single channel permeability is approximately 3×10^9 water molecules per subunit per second (which is $8.97 \times 10^{-14} \text{ cm}^3/\text{s}$). Thus, we used the symbol “~” to be consistent with the original paper we cited. However, in order to make the ~ notation less awkward we have changed the text to read “ $\sim 9 \times 10^{-14} \text{ cm}^3/\text{s}$ ”.

(3) Line 286: “(or glycine, glucose, and sucrose)” Parentheses are confusing about meaning.

We agree with the reviewer. The sentence has been rewritten as follows:

“For WT OmpF, the light scattering intensity decreased at the second stage when WT OmpF reconstituted liposomes were exposed to NaCl containing hypertonic solutions. Also, the light scattering intensity decreased at the second stage when WT OmpF reconstituted liposomes were exposed to glycine, glucose or sucrose containing hypertonic solutions”

(4) Typos:

- Line 173: “Hydrogen atoms and atoms from residues away”

- Line 199: “We identified were 31 UCD and seven CSD designs”

- Line 215: “Adding up free energy of transfer” (should be free energies)

We thank the reviewer for catching these typos. We have corrected them in the revised manuscript and they read as follows:

Line 173: “Hydrogen atoms from all residues and hetero-atoms from membrane-facing residues were excluded from this checklist.”

Line 199: “We identified 31 UCD and seven CSD designs.”

Line 215: “The respective hydrophobicities were computed by adding up the transfer free energy changes ($\Delta G_{\text{transfer}}^{\text{water} \rightarrow \text{ethanol}}$) (from water to ethanol) of each of the amino acid side chains that constitute the inner and outer pore wall.”

Reviewer #3 (Remarks to the Author):

Major comments

1. **Line 122** – based on single channel permeability measurements, it would appear that the fold enhancement is more in the one order of magnitude range, not two.

The reviewer is correct in pointing out this error. It has been corrected in the revised manuscript.

2. **Line 206** – which OCD design is being referred to here? TFPhe or TFTrp or some other Trp/Phe pore ordering? This information would be useful. Additionally, it will help to clarify that the pore is elliptical and the dimensions refer to the major & minor axes.

We have observed a similar effect for both TFPhe and TFTrp as both of Phe and Trp are large hydrophobic groups. The effect is more pronounced in TFTrp (see left panel of Figure 2) as Trp sidechain is somewhat larger.

3. **Lines 220-233** – I am having some trouble understanding this section (perhaps I am missing some thing). I expect the TFTrp OCD design to have the highest hydrophobicity, which is consistent with what the authors claim in line 231-232. However, line 221 asserts that CSD has greater hydrophobicity compared to OCD. Why would this be the case? With 25 Phe(s) or 25 Trp(s) or 25 Trp/Phe combinations, shouldn't OCD hydrophobicity surpass that of CSD? Also, lines 232-233 seem to attribute increased hydrophobicity in OCD to steric clashes between adjacent Trp(s) or Phe(s). I understand that steric clashes can cause lumen widening – but how do they contribute to increasing hydrophobicity?

We understand the reviewer's concern regarding the anomaly between expected and observed hydrophobicity scores of the OCD-TFTrp and CSD pore designs. The OCD-TFTrp indeed has the highest number of hydrophobic residues (with 25 residues altered to Trp at the pore constriction region). However, as we have pointed out in Figure 2, contiguous Trp residues result in steric clashes which causes most of the Trp side chains to face away from the pore lumen. Consequently, these clashing Trp side chains are seen as OmpF surface (or membrane-facing) residues and hence (1) do not contribute towards reducing the pore size, and (2) lead to increased outer pore wall hydrophobicity. However, for the CSD case, this does not happen, as all the thirteen residues at the pore constriction, that have been altered to hydrophobic ones, have their side chains facing the pore lumen, thereby yielding a higher inner pore wall hydrophobicity than OCD-TFTrp (see Table 1). Finally, in lines 232-233, we have tried to re-affirm the same by stating that clashing Trp residues in OCD-TFTrp become membrane facing and thus result in highest outer-pore wall hydrophobicity score (see Table 1). This is now better explained in the revised manuscript.

4. **Lines 256-258** – The UCD design has a smaller pore than the CSD design by ~ 3%. However, from Figure 3b, the smallest pore size bin does not include any UCD design (only a CSD). It seems a bit difficult to reconcile this distribution with the fact that UCD makes provision for the narrowest pore. Could the authors shed some light on this?

We thank the reviewer for pointing this out. This is a mistake in the color coding of Figure 3b. The 2-3 Å and 3-4 Å bins should have their blue and light green bars interchanged. We have reflected this with the new Figure 3b in the updated manuscript.

5. **Line 262** – To what extent are the stopped flow measurements affected by polydispersity in the reconstituted vesicles? Although Prof. Kumar is an expert in this area, some discussion on this would benefit readers who are new to this experimental technique.

High polydispersity of reconstituted vesicles (PDI >0.2) will lead to noisier stopped flow signals and influence the accuracy of fitting the stopped flow light scattering intensity changes. In a previous study we conducted an

extensive analysis of challenges with signal to noise ratios obtained in stopped flow light scattering studies and concluded that when the signal to noise ratio (SNR) is lower than 10, it leads to poor fitting results¹⁹ for permeability calculations. In this paper, the stopped flow curves we used for the permeability calculations have signal to noise ratios that are higher than 50. One way to ensure low polydispersity of reconstituted vesicles (PDI < 0.2) is to extrude the vesicles through 100nm or 200nm track etched membranes and this is what was done in this work. This clarification is now part of the revised manuscript.

“All experiments were conducted with proteoliposomes with a polydispersity index of <0.2 leading to a signal to noise ratio of > 50 in the stopped flow curves that were used for permeability calculation. This is expected to provide high reliability in terms of the calculated parameters from this experiment¹⁹”

6. **Lines 282-284** – *Is there a typing error? If the solute size is larger than the pore size, I would expect the solute to be rejected and not subject to an “inflow of water and solutes into vesicles” as claimed in the manuscript. Please correct/clarify.*

We thank the reviewer for kindly pointing out the typo. We corrected it as following:

“When the solute molecular size is larger than the porin pore size (Figure 4b, solute exclusion model), the water continues flowing outward from the vesicles to reach the equilibrium state of the osmotic pressure dictated by the solute concentration, and the light scattering intensity levels off as measurement time increases.”

7. **Figure 4c** – *the stopped flow data for glycine & glucose appear noisier than corresponding data for sucrose and PEG600, although the trends are still pretty clear. I recommend that the authors provide some explanation to account for the different variabilities in these data sets. Could this be a result of different PDI(s) of various batches of reconstituted vesicles used in the experiments?*

We thank the reviewer for this comment and astute observation. The solute rejection results shown in **Figure 4c** was using the reconstituted vesicles with the same PDI (PDI = 0.11) which is smaller than the prescribed 0.2, thus based on our understanding of the system, PDI may not be a main cause for the noisier stopped flow data for glycine & glucose in **Figure 4c**. Nevertheless, we further conducted a number of new experiments to test this behavior (see Figure below), and we find that the reviewer’s observation is accurate that glycine and glucose provide noisier data than NaCl and PEG 600 on the same batch of vesicles. We have included all the new curves and their values in the SI and updated related numbers in the manuscript – these do not change any conclusions or interpretations but provide more confidence in our values.

We think that one possible reason for the noisier stopped flow data for glycine & glucose in **Figure 4c** may be related to refractive index changes with concentration during the course of the stopped flow data collection. The refractive index change with solute concentration change (dn/dc) is different for sucrose (dn/dc is 0.053)²⁰, glucose (dn/dc is 0.021)²¹ and glycine (dn/dc is 0.011)²². Since stopped flow light scattering experiment measures light scattering change, the differences in the refractive index change as solute concentration change (dn/dc) for different solutes may influence the noise level of the data in a complex non-intuitive manner. Further studies will need to be conducted to further elucidate the true reason behind this subtle difference in scattering behavior.

8. **Lines 302-303** – the stopped flow experiments are capped at the 5 second time point in Figs. 4c & d. Have the authors looked at longer times? Is it possible that tight mutants such as UCD allow solute leakage over time? If true, exclusion benefits of UCD would only be useful over short time-scales. I recommend that the authors address this potential issue by including longer time-scale experiments for NaCl exclusion by UCD. Additionally, it would be useful if the authors could compare their stopped flow time scales with typical times that might be employed in a realistic (i.e., pilot scale) membrane desalination scenario.

In response to the reviewer's request, we performed a 30s stopped flow experiments for both UCD incorporated liposomes and the control lipid vesicles (with no protein incorporated). In this experiment, UCD incorporated liposomes were measured at the same osmolarity difference conditions (10mM) while control lipid vesicles were measured at 10 times higher osmolarity difference condition (100mM) to get high quality data for control lipid vesicles and to evaluate them at similar shrinking rates. For both UCD incorporated liposomes and control lipid vesicles, we observed light scattering intensity decrease at long measurement times, which may due to the over shrinkage of vesicles when the flux is high at the beginning and then a compensation. We have seen such a phenomenon with higher permeability channels in the past at long time scales and hope to explore this further in future studies. We think that for the current study the 5s benchmark is appropriate to look at leakage, particularly because we can compare trends between different mutants and with simulations.

As for comparison to practical pilot scale experiments, there needs to be macro-scale membranes that such tests are done on and we do not think it is practical to connect detailed stopped flow results (other than simple permeability values) to what could be expected at higher orders of magnitude. This is much further along the line but we are making progress towards this.

9. **Line 358** – *Is the 1 protein trimer/micelle assumption valid for OmpF, which has a different size and multimerization propensity compared to AQP?*

This is an excellent observation. Several previous studies have shown that OmpF exists as a native trimer in detergent micelles and in lipid membranes²³. Thus, to confirm the one protein trimer/micelle assumption, we performed Dynamic Light Scattering (DLS) size measurement for both the OmpF micelle samples (Figure A) solubilized from proteoliposomes that were created during the FCS measurement and the OmpF protein solution (Figure B). For the OmpF micelles samples, DLS measurement showed a peak at 7.7nm, which is similar (within the range of error of the DLS measurements) to the 8.2nm intensity peak shown in OmpF protein solution samples (the higher peak values seen are from aggregation of protein in this particular solution). In addition, we also calculated the diameter of OmpF trimer based on OmpF trimer molecular weight and average protein density (1.35g/cm³)²⁴ – this calculated diameter was 6.4nm. Considering that this trimer will be coated with a layer of detergent micelles, it is close to the values measured using DLS. Thus, we submit that the assumption that one protein trimer/micelle is an accurate assumption.

10. **General comment** – *While the work is truly exciting, it could be strengthened by including deeper analyses of the key benefits of these engineered OmpF variants relative to AQP1 or RsAqpZ, perhaps in the context of membrane purifications. This would be in addition to the expanded solute selectivity benefit that the authors talk about in the Discussion section. For instance, is faster flux really critical for membrane applications? What kind of techno-economic (and other) benefits will a 40-fold faster channel confer relative to current aquaporin-based approaches? Furthermore, in the introduction, the authors allude to poor stability of AQP1 as being a limitation. Can they provide some experimental evidence of enhanced stability of the designed OmpF mutants relative to AQP1 for long term or repeated use? I am afraid that the hydrophobic residues in the channel pore could lead to an increased propensity for the pore to “collapse”? Have the authors performed experiments to exclude this possibility?*

We thank the reviewer for recognizing the innovative nature of the work. We agree that a more expanded discussion would be helpful for readers. We have included a deeper analysis of the benefits of OmpF variants compared to AQPs in the text now, including two of three benefits listed below (the third is in the process of being written up for a follow-up publication). There are three key benefits to using OmpF and OmpF mutants when compared to AQPs

1. *AQPs are overdesigned for just salt rejection.* AQPs are designed for high water permeability while rejecting all solutes including protons. The requirement to reject protons imposes the need to have hydrogen bonding between translocating water molecules and the pore wall. Recent studies have indicated that an ideal water-conducting pore could transport water more efficiently if all hydrogen bonding between waters and the central section of the pore are eliminated²⁵ (Horner *et al.*, Science Advances 2015). Changing these residues (especially the key conserved residues of the NPA motif) in AQPs leads to lowered AQP permeability²⁶ (see for example Yong *et al.*, IUBMB Life, 61(6): 651–657, 2009). OmpF provides an excellent platform to tune (or potentially completely eliminate) hydrogen bonding.

2. *A higher water permeability membrane resulting from the proposed OmpF mutants could have benefit for specific applications* where: 1) space is at a premium such as small systems on space missions, desalination plants in crowded coastal cities, 2) where RO operation is still far from approaching the thermodynamic minimum for separations such as in brackish water desalination and wastewater reuse.

3. The OmpF trimer is solvent stable for short amounts of time allowing for solvent casting of membranes when compared to AQPs. We have recently conducted these studies that show functional incorporation of OmpF into polymers in the presence of a chloroform/methanol mixture. Use of solvents allow simple fabrication of membranes by solvent casting which is the default method for making membranes.

Longer-term studies will be needed to address the question regarding long time scale collapse of the OmpF channel with enhanced hydrophobic interiors. These studies will need to be conducted with macro scale membranes built around such channels in order to truly assess if these channels are stable in membrane matrices and do not collapse onto themselves. We are currently working on such membranes and these will be subjects for future publications.

The following paragraph was inserted (along with the aforementioned cited appropriately) into the discussion section of the paper in response to this comment:

“The use of OmpF provides two distinct advantages over the use of AQPs for envisioned applications in aqueous separations. First, AQPs are arguably overdesigned for water desalination as they remove protons along with other monovalent and divalent ions while OmpF can be designed to pass protons while removing other ions. The requirement to reject protons imposes the need to have hydrogen bonding between translocating water molecules and the pore wall. Recent studies have indicated that an ideal water-conducting pore could transport water more efficiently if all hydrogen bonding between waters and the central section of the pore are eliminated²⁵ (Horner *et al.*, Science Advances, 2015). Changing these residues (especially the key conserved residues of the NPA motif) in AQPs leads to lowered AQP permeability²⁶ (see for example Yong *et al.*, IUBMB Life, 2009). OmpF provides an excellent platform to tune (or potentially completely eliminate) hydrogen bonding. Second, the higher permeability of OmpF over AQPs can be advantageous for water purification and desalination in specific instances where space is at a premium or where energy savings can be substantial. Ultrapermeable membranes, such as those based on OmpF, with high salt rejection appropriate for RO have the potential to substantially reduce energy (~45 %) or plant infrastructure (pressure vessels, up to 65%) in low salinity streams⁶ such as brackish water desalination and water reuse. The energy advantage is significantly lower for high salinity seawater applications (15% less energy) but the plant size can be reduced by 44%. Sub-nm pore size membranes for nanofiltration (NF) and ultrafiltration (UF) have diverse applications in water treatment, food production and processing, and energy applications which will also benefit from energy and capital cost reduction.”

Minor Comments

1. **Line 74** – please include a relevant reference

An appropriate reference has been inserted¹⁹ (Grzelakowski *et al.* Journal of Membrane Sciences, 2015)

2. **Line 77** – *Could the authors furnish a reference that describes AQP1 stability as an issue that complicates membrane insertion or long-term use? If ref. 35 happens to be the relevant reference, this is not clear from the bibliography.*

Ref 35 is indeed an important reference describing the challenges of AQP insertion into new matrices and the need to be able to match the hydrophobicity of outer pore wall residues of membrane proteins to enable insertion into specific membranes. We have shown that OmpF outer surface can be changed but such a demonstration has not been shown for AQP. Another reference²⁷ we have added is: To and Torres, Membranes (Basel). 2015 Sep; 5(3): 352–368.

3. **Line 141** – *using “molecular dynamics simulations”*

The edit has been incorporated.

4. **Line 180** – *“redesigned” what?*

We have edited it to “redesigned pore”.

5. **Figure 2, caption** – *I think the authors meant to say “UCD designs intersperse smaller side chain hydrophobic amino acids”. Is that correct?*

That is correct, we thank the reviewer for pointing this out. The corrected sentence reads: “UCD designs intersperse smaller side chain hydrophobic amino acids”

References

1. Lout, K. L. *et al.* Structural and functional characterization of OmpF porin mutants selected for larger pore size. I. Crystallographic analysis. *J. Biol. Chem.* **271**, 20669–20675 (1996).
2. Kumar, M., Grzelakowski, M., Zilles, J., Clark, M. & Meier, W. Highly permeable polymeric membranes based on the incorporation of the functional water channel protein Aquaporin Z. *Proc. Natl. Acad. Sci.* **104**, 20719–20724 (2007).
3. Wang, H. *et al.* Highly Permeable and Selective Pore-Spanning Biomimetic Membrane Embedded with Aquaporin Z. *Small* **8**, 1185–1190 (2012).
4. Ireta, J., Neugebauer, J. & Scheffler, M. On the accuracy of DFT for describing hydrogen bonds: Dependence on the bond directionality. *J. Phys. Chem. A* **108**, 5692–5698 (2004).
5. Durrant, J. D. & McCammon, J. A. HBonanza: A computer algorithm for molecular-dynamics-trajectory hydrogen-bond analysis. *J. Mol. Graph. Model.* **31**, 5–9 (2011).
6. L.J. ZEMAN, ZYDNEY, A. L. & Dekker, M. Microfiltration and Ultrafiltration - Principles and Applications. *Chemie Ing. Tech.* 1479 (1996). doi:10.1002/cite.330691022
7. Erbakan, M. *et al.* Molecular Cloning, Overexpression and Characterization of a Novel Water Channel Protein from Rhodobacter sphaeroides. *PLoS One* **9**, e86830 (2014).

8. Bocquet, N. *et al.* X-ray structure of a pentameric ligand-gated ion channel in an apparently open conformation. *Nature* **457**, 111–114 (2009).
9. Hibbs, R. E. & Gouaux, E. Principles of activation and permeation in an anion-selective Cys-loop receptor. *Nature* **474**, 54–60 (2011).
10. Hilf, R. J. C. & Dutzler, R. X-ray structure of a prokaryotic pentameric ligand-gated ion channel. *Nature* **452**, 375–379 (2008).
11. Miyazawa, A., Fujiyoshi, Y. & Unwin, N. Structure and gating mechanism of the acetylcholine receptor pore. *Nature* **423**, 949–955 (2003).
12. Liu, Z., Ghai, I., Winterhalter, M. & Schwaneberg, U. Engineering Enhanced Pore Sizes Using FhuA Δ 1-160 from *E. coli* Outer Membrane as Template. *ACS Sensors* **2**, 1619–1626 (2017).
13. Fairman, J. W., Noinaj, N. & Buchanan, S. K. The structural biology of β -barrel membrane proteins: A summary of recent reports. *Current Opinion in Structural Biology* **21**, 523–531 (2011).
14. Pantazes, R. J., Grisewood, M. J., Li, T., Gifford, N. P. & Maranas, C. D. The Iterative Protein Redesign and Optimization (IPRO) suite of programs. *J. Comput. Chem.* **36**, 251–263 (2015).
15. Székely, G., Bandarra, J., Heggie, W., Sellergren, B. & Ferreira, F. C. Organic solvent nanofiltration: A platform for removal of genotoxins from active pharmaceutical ingredients. *J. Memb. Sci.* **381**, 21–33 (2011).
16. Perunov, N. & England, J. L. Quantitative theory of hydrophobic effect as a driving force of protein structure. *Protein Sci.* **23**, 387–399 (2014).
17. Kister, A. E. & Phillips, J. C. A stringent test for hydrophobicity scales: two proteins with 88% sequence identity but different structure and function. *Proc. Natl. Acad. Sci. U. S. A.* **105**, 9233–9237 (2008).
18. Kozono, D., Yasui, M., King, L. S. & Agre, P. Aquaporin water channels: Atomic structure and molecular dynamics meet clinical medicine. *Journal of Clinical Investigation* **109**, 1395–1399 (2002).
19. Grzelakowski, M., Cherenet, M. F., Shen, Y. xiao & Kumar, M. A framework for accurate evaluation of the promise of aquaporin based biomimetic membranes. *J. Memb. Sci.* **479**, 223–231 (2015).
20. <http://www.refractometer.pl/refraction-datasheet-sucrose>. Refractive indices of sucrose solutions at 20 °C, 589.29 nm.
21. Yeh, Y.-L. Real-time measurement of glucose concentration and average refractive index using a laser interferometer. *Opt. Lasers Eng.* **46**, 666–670 (2008).
22. Soto, A., Arce, A. & Khoshkbarchi, M. K. Experimental data and modelling of apparent molar volumes, isentropic compressibilities and refractive indices in aqueous solutions of glycine+NaCl. *Biophys. Chem.* **74**, 165–173 (1998).
23. Cowan, S. W. *et al.* Crystal structures explain functional properties of two *E. coli* porins. *Nature* **358**, 727–733 (1992).
24. Fischer, H., Polikarpov, I. & Craievich, A. F. Average protein density is a molecular-weight-dependent function. *Protein Sci.* **13**, 2825–2828 (2009).
25. Horner, A. *et al.* The mobility of single-file water molecules is governed by the number of H-bonds they may form with channel-lining residues. *Sci. Adv.* **1**, e1400083 (2015).
26. Jiang, Y. Expression and functional characterization of NPA motif-null aquaporin-1 mutations. *IUBMB Life* **61**, 651–657 (2009).
27. To, J. & Torres, J. Can stabilization and inhibition of aquaporins contribute to future development of biomimetic membranes? *Membranes* **5**, 352–368 (2015).

REVIEWERS' COMMENTS:

Reviewer #1 (Remarks to the Author):

The author addressed all my requests and comments.

Reviewer #2 (Remarks to the Author):

The authors have thoroughly addressed all concerns. Thank you to the authors for detailed consideration of all concerns.